# Analyses of Genetic Diversity in the Endangered “Berrenda” Spanish Cattle Breeds Using Pedigree Data

**DOI:** 10.3390/ani12030249

**Published:** 2022-01-20

**Authors:** Rafael González-Cano, Ana González-Martínez, María Eva Muñoz-Mejías, Pablo Valera, Evangelina Rodero

**Affiliations:** 1CERSYRA-IRIAF, Avenida del Vino 10, 048619 Valdepeñas Ciudad Real, Spain; rgcano@jccm.es; 2Department of Animal Production, Faculty of Veterinary Sciences, University of Cordoba, 14071 Córdoba, Spain; agmartinez@uco.es; 3Department of Animal Pathology, Animal Production, Bromathology and Food Technology, University of Las Palmas de Gran Canaria, 35001 Las Palmas de Gran Canaria, Spain; mariaeva.munoz@ulpgc.es; 4Gescan, Gestión de Programas de Cría, 35017 Las Palmas de Gran Canaria, Spain; directortecnico@gescansl.com

**Keywords:** inbreeding, genealogical data, conservation, native resources, genetic contribution

## Abstract

**Simple Summary:**

The two “berrenda” cattle breeds are important for the conservation of livestock genetic diversity in Spain. They have a great phenotypic and genotypic uniqueness and both of them are important from the cultural and the tourist perspectives. They also contribute to the conservation of the traditional “Dehesa” ecosystem. Both breeds are considered as endangered ones, but their genealogies have never been used for quantifying the risk status of their populations. The aim of this work was to monitor the structure of the “Berrenda en Negro” and the “Berrenda en Colorado” populations, their inbreeding rate and some other parameters that could be useful to prevent losses in their genetic diversity and to conduct and analyze the effect of the conservation programs developed by ANABE Breeders Association and finally, as a tool to implement some selective measures. We found that both “berrenda” cattle breeds retain a huge genetic variability from their founders’ populations, although they have been affected by a shallow depth in their pedigrees; as a consequence, we suggest increasing exchanges of breeding animals among herds, more specifically in the case of the “Berrenda en Negro” breed.

**Abstract:**

Pedigree analyses of two endangered cattle breeds were performed in order to study the structure and the genetic variability in their populations. Pedigree data were analyzed from 12,057 individuals belonging to the “Berrenda en Negro” cattle breed (BN) and 20,389 individuals belonging to the “Berrenda en Colorado” cattle breed (BC) that were born between 1983 and 2020. BN and BC reference populations (RP) were set up by 2300 and 3988 animals, respectively. The generation interval in BN and BC reference populations was equal to 6.50 and 6.92 years, respectively. The pedigree completeness level was 82.76% in BN and 79.57% in BC. The inbreeding rates were 4.5% in BN and 3.4% in BC, respectively. The relationship among animals when they were born in different herds was 1.8% in BN and 5% in BC; these values increased to 8.5% and 7.7%, respectively when comparing animals that were born in the same herd. The effective number of founding herds was 23.9 in BN and 60.9 in BC. Number of ancestors needed to explain 50% of genes pool in the whole population was 50 and 101, in BN and in BC, respectively. The effective population size based on co-ancestries was 92.28 in BN and 169.92 in BC. The genetic variability has been maintained in both populations over time and the results of this study suggest that measures to promote the conservation of the genetic variability in these two breeds would go through for the exchange of breeding animals among farms and for monitoring the genetic contributions before implementing any selective action.

## 1. Introduction

Farm Animal Genetic Resources in Spain possess a remarkable variability as a consequence of the country unique geographical location between Africa and Europe. In fact, 46 autochthonous cattle breeds are officially recognized [1]. Among all these cattle breeds, the Berrenda en Negro (BN) and the Berrenda en Colorado (BC) breeds were officially considered as breeds in danger of extinction in 1979 by the Spanish Minister of Agriculture [2]. Currently, they still have a scarce number of registered breeding animals: 2584 in BN and 4186 in BC, respectively [3].

Although the origin of the “berrenda” breeds is considered as ancestral and previous to the Columbian era [4], the first references, published in the 19th century, considered both breeds as a unique racial group [5]. Mitochondrial mt-DNA analyses have shown that both ‘“berrenda’” breeds have different origins [6] and it is being estimated that the separation between them only happened 180 years ago [7]. Consequently, they are officially considered as two distinct breeds, they are subjected to their own specific breeding programs and they have separated studbooks [8] that are managed by ANABE, Group of Berrenda en Negro and Berrenda en Colorado Cattle Breeders Associations [9].

Regarding their phenotypes (Figure 1), both above-mentioned breeds are called “Berrenda” for their coat color pattern. “Berrenda” means “spotted coat” in Spanish language. Although, their morphology [10] and coat color pattern [11,12] are quite similar, they mainly differ in the color of the spots. So, while BN shows spots in black, BC is red spotted. Similarly to other local breeds [13], coat color and spot patterns in both “berrenda” cattle breeds are relevant for their appearance and for the differentiation between them, especially when they are used in traditional events. In order to achieve the fixation of black coat color in BN and to avoid births of animals with different coat color that could not be admitted as BN breeding animals, BN studbook in combination with genotyping of MC1R locus have been used [14].

Berrenda en Negro and Berrenda en Colorado herds are geographically dispersed in “Dehesa” ecosystems, located in southern and middle Iberian Peninsula [15] (Appendix A). “Dehesa” is the Iberian traditional ecosystem that is part of the Natura Network 2000 because of its high natural value [16]. These cattle are reared under extensive conditions, grazing natural meadows and playing an important role in maintaining the “Dehesa” ecosystemic balance by contributing to the maintenance of its vegetation covering [17,18] and its socioeconomic competitiveness [19,20]. Under these breeding conditions and due to their high rusticity, “berrenda” females are excellent mothers for rearing F1 cross-breeding calves, after mating with Charolais or Limousin sires.

In addition, and due to their natural character and behavior, these breeds can be easily taught, what makes them to be of special interest among all the others Spanish autochthonous cattle breeds [5]. Oxen of both “berrenda” breeds are used as draught animals in religious and popular events, such as pilgrimages and festivals. In addition, they are the only ones that are used in the management of bull-fighting herds in the countryside or in popular “running of bulls”, such as “San Fermín” festivals in Pamplona [5] (Appendix A).

In addition, Cañón et al. [21] have reported that these two breeds have remarkably contributed to the global genetic diversity among the different Spanish cattle breeds. In fact, they are quite interesting because they represent the phylogenetic relationships between the African cattle breeds and the Southern Spanish autochthonous ones [6,22,23].

In summary, these are two autochthonous cattle breeds of high conservation interest, due to multiple reasons: historical, religious and cultural, linked to local traditional events that are important for the socioeconomic development of the regions where they are reared, mainly linked to tourism activities. They also contribute to the sustainable use and maintenance of the “Dehesa” unique ecosystems, increasing their environmental and economic value by producing meats of differentiated quality, linked to specific type of animals that are reared to be slaughtered (oxen). In addition, as ancestral breeds, they are also of special interest from the genetic point of view. The prospects for the development and conservation of these breeds rely on the ability of consumers and institutions to recognize these values and on the potential of these breeds to face future challenges related to possible changes in consumers’ preferences, to sustainable economy of the territories and to adaptation to climate changes. Market demands could only be met if the implementation of breeding programs developed measures addressed to the prevention of inbreeding. Nowadays, the main objectives of the “berrenda” breeds breeding programs are the conservation of the populations (mainly at in vivo and in situ level) and the maintenance of their genetic variability. Currently, the selective activities carried out by ANABE are focused on not using carriers of phenotypic or genetic disorders, such as Robertsonian chromosomal translocation t1:29, as breeding animals [15] and also avoiding undesirable mating between related animals, in order, to avoid inbreeding depression. Moreover, ANABE is planning to implement a program in coming years addressed to improve females rearing skills by analyzing calves’ growing rates and by calculating their estimated breeding value (EBV) using BLUP, the genetic estimation method according to the algorithm by Meuwissen et al. [24]. Semen collection and freezing (for in vitro conservation purposes) are being carried out from those bulls with the highest genetic variability, no t1:29 carriers and in some cases, with their corresponding EBVs. In addition, ANABE is planning to start an artificial insemination (AI) program in 2022 addressed to preserve Genetic Variability.

Monitoring genetic diversity and monitoring genetic trends in animal populations are the bases for the implementation of any conservation program, as well as a suitable selection of the offspring, in order to reach a sustainable livestock farming [25]. The stochastic demographic risks and the genetic threats, such as inbreeding depression, fixation of deleterious mutations or losses in adaptive potential, are interconnected [26,27].

Traditionally, performing pedigree analyses was the first step towards the characterization of the genetic diversity within a population but not many works of this kind related to small populations have been published yet [28]. On the other hand, these analyses would also provide a better knowledge of the history of the populations making possible to detect different events from the past that could have affected them, such as the identification of founders’ population or the identification of bottlenecks [29]. Certainly for these purposes, information provided by modern genomic methods is quite better than information provided by pedigree analyses [30,31,32]. Genomic data would allow for a more accurate assessment addressed to determine different parameters about the structure of the population, such as the Effective population size or the Inbreeding rate [26,27,33], with the added advantage that when using genomic parameters, we would be describing the realized genetic variation and not the expected one. Genomic tools are also quite effective for the conservation of diversity in livestock breeds when they are applied to the study of the genomic linkage disequilibrium due to genetic (selection, mutation, genetic drift, non-random mating, etc.) and non-genetic forces [34]. Genomic analyses demand high financial cost, so the usage of genomic data is not so common in small livestock breeds and analyses based on pedigree data are still the preferential option for monitoring livestock conservation programs, supported by a combination of traditional [35,36,37] and genomic selection strategies [26,38].

Although the first registrations of animals in BN and BC studbooks took place in 2001, the traditional rearing system under extensive conditions, in combination with the insufficient control of mating by breeders made quite difficult to correctly assign the paternity to the offspring, what negatively affected to the amount of information provided by pedigrees, in the first stages of the breeding program [35]. Since then, to now, 232 BC herds and 152 BN herds have been registering animals in their corresponding studbooks, reaching 12,057 and 20,389 registered animals in BN and BC, respectively [3].

In the case of “berrenda” cattle breeds, genetic diversity and paternity of the offspring are annually monitored by analyzing DNA microsatellites, according to FAO recommendations [39]. Although the neutral genetic variability is keeping reasonably high in both “berrenda” breeds, apparently it is decreasing in the last 6 years, as a consequence of the implementation of the above-mentioned actions [40], what could make necessary to complete the studies on genetic variability by using a different methodology.

We can find some studies on main Spanish autochthonous cattle breeds based on pedigree data [31,41,42,43,44], but not many among them have been focused on threatened breeds [42,43,45,46] and they have been mainly aimed at the study of some rare cattle breeds located in Northern Iberian Peninsula (Galicia [43], Asturias and Catalonia [41,45]), or in the Canary Islands [46], but never based on rare breeds located in the “Dehesa” ecosystems.

As a contribution for the development of the conservation programs in both “berrenda” cattle breeds, this work aims to study the current genetic variability and the structure of BN and BC populations based on thorough analyses of their corresponding pedigrees, by quantifying the changes that happened over the time and comparing them to other different autochthonous cattle breeds that have also been traditionally reared in the “Dehesa” ecosystems, following a quite similar farming system.

To be more precise, the objective of this work are: (i) characterizing the structure of BN and BC populations by ages and sex and type of herd; (ii) identifying the genetic variability level and the threat of inbreeding in both populations; (iii) identifying the genes origin in both populations. It is also intended to know the possible loss of genetic.

This information will be quite useful to establish the most suitable strategies to be implemented in the conservation and selection programs of these valuable genetic resources in risk of extinction.

## 2. Materials and Methods

### 2.1. Data and Reference Population

Based on the records from BC and BN breed’ studbooks and after removing all those founders without descendants, the analyses were performed over the BN and BC breeds whole populations.

Subsequently, the reference populations were made up of live animals born during the last 7 years (from 2013 to 2020). According to that, BN breed reference population was composed by 2300 animals and BN breed reference population by 3988 individuals. This period has been established, according to the average generation interval from live animals in the whole populations.

### 2.2. Calculated Parameters

For descriptive purposes about the structure of both populations, we have designed on Microsoft Excel 365^®^ for Windows, US, some tables and figures, based on different values: number of animals by sex, number of animals by year of birth, as well as a description of the structure of the populations according to the age of the registered animals in the studbooks that are still alive and the number of registered descendants.

Furthermore, to describe the structure of the studbooks and the integrity of the information than they contain, we analyzed the next parameters in both breeds:(i)Generation interval: It is defined as the average age of parents (male or female) when its replacement is born [47]; it is calculated considering the four possible paths: sire-daughter, sire-son, dam-daughter and dam-calf;(ii)Proportion of present ancestors per generation: It was estimated in order to analyze the pedigree completeness level [48];(iii)Pedigree depth: It is calculated by considering the equivalent number of discreet generations equals to ${\left(1/2\right)}^{n}$, where *n* is equal to the number of generations between the individual and its known ancestor; the individuals without known ancestors were assigned to the base generation [49]. The maximum number and complete generations of the pedigree have also been calculated. On the other side, the Inbreeding Coefficient (*F*) and the Average Relationship Coefficient (*AR*) have been calculated to analyze the inbreeding and the genetic weight of every individual over the whole population. *F* is defined as the likelihood that an individual carries two identical genes by descent. *AR* is defined as the average genetic weight of every individual over the whole population and it is equal to the average coancestry coefficient of every individual related to the other members of the population [42,50].

Moreover, the Inbreeding rate per equivalent generation (*∆F*) and the Inbreeding after 10 years (*F_10_*) and after 50 years (*F_50_*) were calculated according to the methodology proposed by Gutiérrez and Goyache [51], as follows: $\Delta F_{i}=1-\sqrt[{t-1}]{1-F_{i}}$; where F*_i_* represents the individual inbreeding and t is the equivalent per complete generation.

In addition, the Effective Population Size (*N_e_*) and the Effective Population Size after 10 and 50 years (*N_e10_* and *N_e50_*) were calculated based on the increasing of the individual inbreeding in the animals included in the Reference Population, according to the *N_e_* value obtained from the approximation by Gutiérrez et al. [52,53], that takes into consideration the genetic history of the populations, the size of the founders population and the possible bottlenecks; and that also appears to be more robust than the regression methods applied to deeper pedigrees [54]. Further, these parameters have also been calculated based on the Increasing of Coancestry (*ΔC*), following the methodology proposed by Cervantes et al. [33], according to which: $N_{ec}=\frac{1}{2\Delta C}$. This parameter has been considered more effective for incomplete pedigrees than the previous ones [55].

With the objective of determining the original genes concentration and the genetic variability in both “berrenda” cattle breeds, the following parameters were analyzed:(i)Effective number of herds that are rearing sires performing as grandparents, great-grandparents and great-great-grandparents: It was calculated according to Robertson [56] and it is defined as the inverse of the probability that two animals randomly chosen could belong to the same herd. Robertson’s statistics also allow identifying the herd they belong, depending on the exchange of breeding animals with other herds (nuclear, multiplier, commercial or isolated herds).(ii)Effective number of founders (*f_e_*) that equally contributes and determines the existing genetic diversity in the population: It was calculated according to fe=1∑k=1fqk2, where *q*1 represents the genetic contribution to the population of founder [57].(iii)Effective number of ancestors (*f_a_*): It is defined as the number of ancestors, founders or not, that are needed to explain the population whole genetic variability [58].(iv)Marginal contributions of ancestors *f_a_*/*f_e_* reveals the possible bottlenecks that could have affected the population under study. It also considers the genetic variability provided by an animal that could not be explained by the contribution of its offspring. It was calculated according to Boichard [58].(v)Number of Founder Genome Equivalents (*f_g_*): It can be defined as the number of founders that would lead to a similar genetic diversity in the population under study if the founders were equally represented and with no loss of alleles [59]. The values of this parameter in each “berrenda” cattle breed were calculated as the inverse of twice the average coancestries in the reference populations [60].(vi)Genetic Conservation Index (*GCI*): It is calculated based on the genetic contributions of all the individuals considered as founders (*p_i_*) [4]. According to that, the highest values would be obtained by those individuals retaining a higher number of alleles from the existing ones in the founders’ population: $GCI=\frac{1}{\sum p_{i}^{2}}$.(vii)The amount of genetic diversity (*GD*) in the reference population, accounting for loss of diversity due to genetic drift and the unequal contribution of founder was calculated as follows [61]: GD=1−12fge.

When it is expressed as 1-*GD*, it estimates the loss in the genetic diversity of the population since founders’ generation due to bottlenecks and genetic drift.

The amount of genetic diversity in the reference population accounting for loss of diversity due to the unequal contribution of founders (*GD* *) was calculated as follows [62]: GD ∗=1−12fe.

Similarly, the loss of genetic diversity due to the unequal contributions of founders was estimated by 1 − *GD* *. The difference between *GD* * and *GD* estimates the loss of diversity by genetic drift accumulated over non-founder generations and it is equal to the inverse of 2*N_enf_* [60].
(viii)The degree of genetic differentiation among herds, considered as the contribution level of the populations to the whole genetic variability, were estimated using Wright’s F statistics (1978) according to Caballero and Toro [62] and were adapted to the subpopulation sizes, as specified by Bartolomé et al. [63].

Lastly, all the previous parameters were calculated using ENDOG Software (V. 4.8) and they all have been described in detailed by Gutierrez et al. [64].

## 3. Results and Discussion

### 3.1. Population Structure and Pedigree Completeness Level

Berrenda en Negro and Berrenda en Colorado official studbooks started up in 2005, after the registration of 2651 cows and 523 bulls and 1626 cow and 246 bulls in the BC and BN studbooks, respectively.

According to birth dates (Figure 2), the annual evolution in the number of admissions in the studbooks, showed that it was increasing up to 2012. From that moment, falls in the number of registrations, could be associated on the one side, to the strong dependence of these breeds on EU common agricultural policy subsidies for endangered autochthonous breeds [20,35,65,66] and on the other side, to the effects of the amendments in the Spanish regulation on the registration of new founder males in the studbooks from 2013 onward [8]. Regarding to the number of registered herds, it has not followed the same trend than the number of registrations in the studbooks, showing a decrease since 2009, although the number of animals per herd has remained constant.

Similarly, to many other rustic livestock breeds that are extensively reared in the “Dehesa” ecosystem, the reproductive parameters are not quite good. In the case of BN, the average age of parents when their first offspring is born is 5.51 ± 3.68 years for dams and 3.20 ± 1.44 years for sires. In BC, the obtained values have been quite similar: 5.27 ± 3.40 years for dams and 3.56 ± 1.67 for sires. Although there is a huge variation among animals, the average age of parents when their last offspring is born is 8.63 ± 4.50 years for dams and 5.85 ± 2.71 years for sires belonging to BN and in BC are 8.63 ± 4.48 and 6.48 ± 3.00 years for dams and sires, respectively.

Moreover, the whole proportion of both BN and BC breeding animals that have offspring when they are more than 10 years old was 12% in sires and 36% in dams; and, 10.4% of dams went on giving birth when they were older than 15 years (Figure 3). Age differences between sires and dams also suggested a longer reproductive life in females, especially in BC. The highest age of females was expected, because selection strategies have largely relied on sires. The longest use of bulls as sires could benefit both the conservation and the selection programs, whenever mating between related animals were avoided [67,68].

The average number of descendants registered in the studbooks in the whole BN population was 2.96 ± 2.19 calves per dam and 35.30 ± 42.39 calves per sire. In the case of BC, the data are quite similar to BN, presenting 3.13 ± 2.29 registered descendant per dam and 40.61 ± 48.45 registered descendant per sire. Approximately 30% of BN sires have less than 10 descendants and 24% of them have more than 50 offspring. In the case of BC breed, 22% of the sires have less than 10 descendants and 25% of them have more than 50 descendants (Figure 4a). In the case of BN, 12.49% of the whole offspring are descendant of sires with less than 20 offspring. BC is a little bit more prolific than BN because that value arises up to 9.66% and regarding dams (Figure 4b) 23% of BC have given birth to two calves but only 21% of BN.

There are some important facts that are limiting the increase in the number of breeding animals registered in both “berrenda” cattle breeds; among them, we will stand out the following ones: (i) 50% of dams are crossbred with meat sires in order to get calves with a higher meat yield; (ii) genetic selection against calves carrying t1:29 chromosome disorder; (iii) the lack of fixation of MC1R locus dominant allele E^D^ in BN population, responsible of black coat color in BN what could lead to the feasible birth of red or chestnut coat color calves which cannot be registered in the BN studbook nor used as breeding animals [8] and (iv) the castration of bulls to become oxen.

### 3.2. Studbooks’ Compression and Integrity of the Information

Estimates of the generation interval (*L*) considering both the BN and BC whole populations and the BN and BC reference populations are shown in Table 1; the global average interval between generations in references populations was equal to 6.50 ± 0.10 in the BN and 6.92 ± 0.08 in the BC. Mean *L* values were longer than reported by Cañas-Álvarez et al. [31] about other Spanish cattle breeds raised in the “Dehesa” ecosystem (*L* = 3.7 in Avileña, 6.4 in Morucha and 6.2 in Retinta) but shorter than estimates by Cortés et al. [44] in the whole population of Bull-fighting cattle (L = 7.5). Longer *L* values are due to Farmers’ bias in favour of the reproductive use of certain animals performing specific external traits. Most popular dams and sires might have been used as breeding animals for long periods of time and their progenies would have had a high influence on the next generation [69]. Counting with a higher number of bulls and cows performing as sires and dams, would lead to a progressive increase in the effective population size and might prevent a fast raise of both the inbreeding and the genetic drift in the population [24]. Considering that the main objective in the case of the “berrenda” cattle breeds is the conservation of the populations and not increasing their genetic gain, the most appropriate approach to succeed at the short term, it would be to extend both the generation interval and the reproductive use of breeding animals.

However, it is necessary to be careful in the control of the mating strategies; on the contrary, there would be a high risk of mating between animals belonging to the same lineage, what would lead to an increase of inbreeding [69]. High *L* values would lead to lower genetic progress and lower genetic gains (in terms of selection of desirable traits) [70]. Genomic selection allows carrying out selective actions addressed to improve some traits of economic interest on the basis of small reference population sizes and moderately long generation intervals, without finding significant differences in the expression of Bulmer’s effect between genomic selection and other traditional selection approaches based on BLUP methodology [71].

Generation intervals of the four different pathways were not always similar as there is a certain unbalance among the generation interval values in both “berrenda” breeds. A slightly higher generation interval was observed in the Dam-Offspring pathways than in the Sire-Offspring ones, what could be determined by replacing sires at an earlier age than dams (Table 1).

Additionally, the BN generation interval stands out in the Sire-Son pathway (4.92 ± 0.17) in comparison to the Dam-Daughter (7.09 ± 0.09) one. According to Gicquel et al. [35], the existing unbalance among the four different pathways could bias the number of equivalent generations (*EqG*), what could explain the higher value of this parameter in BN (Table 2), despite the fact that the average parental age at birth of their offspring in the four possible pathways was lower in BN than in BC.

These results (longer *L* values for dams than for sires) are coincident to those reported by Cañas-Álvarez et al. [31] about seven Spanish cattle breeds and to the majority of studies performed on cattle populations located in other geographical scenarios, such as Limousin and Charolais cattle located in Italy [72], the Maremana breed [36] and the Sahival population located in Kenya [73]. On the contrary, they are opposed to the results reported by Santana et al. [67,68] in seven Hindu-Brazilian local cattle breeds and to the reports about Holstein and Jersey cattle from Canada, after 30 years of genetic selection [26].

As a general rule, the evaluation of the population structure, based on relative information, trends to be much more efficient and precise when the number of maximum generations, number of complete generations and number of equivalent generations are high. Usually, during the first steps of every conservation and reproduction program there is an incomplete knowledge of the relationships among the animals that are registered in the studbook, what could lead to an underestimation of the average number of equivalent generations [74]. Considering “berrenda” cattle breeds, *EqG* values were quite low (1.61 in BN and 1.54 in BC), despite counting with a high number of registrations in the studbooks (12,057 in BN and 20,389 in BC). Increase in the number of known generations would provide a higher certainty in the estimations of the population parameters [75].

With low pedigree depths, both in BN and in BC, there will be a trend to overestimate the consanguinity rate and to underestimate the effective population size. Nevertheless, our *EqG* results are quite similar to those *EqG* values initially used by Gutiérrez et al. [42] in the Spanish Alistana, Asturiana de las Montañas (1.6) and Sayaguesa breeds (1.7). Among all the Spanish meat cattle breeds traditionally located in the “Dehesa” ecosytem, this parameter varied from 2.1 in the Morucha breed to 3.9 in the Avileña-Negra Ibérica and the Retinta breeds [31]. Logically, are in contrast to those found in transboundary high selected breeds such as the Charolais (CH) and the Limousin (LM) breeds, which are reared in other Mediterranean areas. *EqG* value was 9.3 in CH and 7.3 in LM, considering the French populations of these two breeds [76]. Recently and referring to Italian CH and LM populations, Fabri et al. [28] found values of 18 and 15, respectively. Pedigree completeness rate was quite high in the first generation of BN and BC whole populations, where the average percentage of known ancestors was 82.76% in BN and 79.57% in BC, respectively. Nevertheless, these percentages are higher in the reference populations, reaching 99.25% in BN and 99.25% in BC, respectively (Figure 5 and Figure 6). However, pedigree completeness rate in the third generation was more moderate than in the latest generations that showed a pedigree completeness rate closed to 40% in both reference populations, slightly better in BN (47.5%) than in BC (37.8%), although it was closed to zero in prior animals to the fourth generation.

Currently, and according to Regulation (EU) 2016/1012 [77], females with unknown ancestors might be registered in the auxiliary section of the herd-books. In the case of the “berrenda” breeds, pedigrees of these animals are not always complete and some of them could be considered as founder animals. However, the last generations that have been registered in the BN and BC herd books have shown a better quality because a very higher degree of completeness (Figure 5 and Figure 6). Results showed that in the last year, pedigree quality has experienced an improvement after the closure of the studbook males’ foundational section, establishing that only a bull can be considered as a sire, when its parents and grandparents are known and registered in the corresponding studbook. Therefore, reference populations will provide best values and less biased thanks to the highest quality of the information. However, pedigree completeness rate is quite similar in the four possible pathways and in the generations between dams and sires (Figure 6).

### 3.3. Inbreeding Analysis and Non-Random Mating Rate

In small populations as “berrenda” cattle breeds, phenotypic frequencies change by increasing homozygosity; therefore, reducing genetic inbreeding is the first objective in the management of at-risk populations when inbreeding depression occurs. It is also very important in breeding programs to give raise to productive offspring [78] and negative effects of inbreeding depression has been shown noticeable when F values are above 20% in beef cattle breeds [79].

Despite not having detected any mating between parents and their offspring, the average percentage of inbred animals (*Inb*) was 63% in BN and 54% in BC (Table 2). The average inbreeding coefficient (*F*) of the animals registered in the BN and BC studbooks was 4.5% and 3.4%, respectively. Low pedigree completeness levels (mostly in the first generations of those pedigrees that apparently are not quite deep) could also lead to an underestimation of the inbreeding levels because descendants from known parents are given an inbreeding value equal to 0, even if they are related to others [54].

The calculated inbreeding rates (*∆F*) in “berrenda” cattle breeds present an increase in the consanguinity per equivalent generation (4.02% in BN and 3.01% in BC) as it is shown in Table 3. In both cases, it is higher than 1.5% what means that consanguinity rates after 50 years are equal to 33.7% in BN and equal to 18.78 in BC. Thinking on the future, *ΔF* and *N_e_* values found for BN and BC exceeded the recommended limits to maintain appropriate levels of genetic variability and to prevent future losses (*ΔF* < 1% and *N_e_* > 50); in any case, BN is in a worse situation than BC. However, we should be aware that pedigree information is still very inconsistent and estimations of *∆F* and individual inbreeding coefficients are very sensitive to the quantity and quality of available pedigree information, in the light of the foregoing [58].

However, more reliably, mean F values obtained in the more complete reference populations amounted to 7% and 5.7% in BN and BC, respectively and they showed to have a high percentage of inbred animals (49.91% in BN and 36.86% in BC references populations) Animals with inbreeding rates above 20% were 14.30% and 9.75%, for BN and BC, respectively, point out that a reduction in the inbreeding rate in both “berrenda” cattle breeds should be necessary. The fact that the average inbreeding values in the current generations of both “berrenda” cattle breeds are more than twice as high as the coancestry values, suggests that related animals have been involved in breeders’ mating plans.

Considering the average inbreeding data from several worldwide cattle breeds reported by Carolino et al. [13], Inbreeding coefficient (*F*) in the BN and BC reference populations were quite similar to those from other local breeds that were reared in extensive farming systems (Dehesa ecosystem) from Portugal (Mertolenga, Alentejana and Preta breeds) and Spain (Avileña Negra Ibérica, Retinta and Morucha breeds and Bullfighting cattle), despite the fact that these last mentioned breeds showed higher pedigree depths (*EqG* > 3) than BN and BC breeds.

In any case much higher than estimates from Charolais and Limousin populations located in France, Sweden, Denmark, Italy and Slovakia that were below or close to 1. If we compare these data to others reported by Cañas-Álvarez et al. [31], regarding different endangered Spanish meat breeds also reared in extensive farming systems, such as Asturiana de las Montañas (*F* = 1.55) or Bruna del Pirineo breeds (*F* = 0.25). BN and BC results are worse but in any case, better than those reported in the Pallaresa breed (*F* = 18.93%) with an effective population size of 4.74 [45]. Inbreeding depression is more pronounced in those breeds that have been rapidly developed by applying inbreeding programs [42], but this is not the case for the ancestral “berrenda” cattle breeds.

When average inbreeding was analyzed at herd level (Figure 7), more than 27% of BN herds and more than 48% of BC herds showed an average inbreeding rate higher than 5%. In any case, males from both breeds are higher inbred than females. Within herd *F* was 7.4% in the BN reference population and 5.5% in the BC reference population, what confirms that mating within herds is mainly carried out between related animals and as a consequence of this, 9.72% of BC herds and 1.65% of BN herds presented inbreeding rates over 20% (Figure 7). It follows that there were great problems to exchange sires among herds and also that it was quite difficult to design mating programs led to minimize the inbreeding rates.

The average relationship coefficients (*AR*) in both “berrenda” cattle breeds (Figure 6) did not show quite high values (Figure 7) and they varied not much through the years (0.95% in BN and 0.40% in BC) (Figure 8), although they increased during the second generation and they followed a slightly increasing trend, considered as not remarkable. When comparing the average relationship between animals born in different herds from the reference populations (Figure 8), the average *AR* values were 1.8% (0.018 ± 0.019) in BN and 5% (0.050 ± 0.062) in BC, increasing to 8.5% in BN and 7.7% in BC when comparing animals born in the same herd. The typical reproductive isolation of BN and BC herds in the “Dehesa” ecosystem, where bulls usually perform as sires in the same herd they have been born and the scarce exchange of animals among herds could explained those results.

If we observed the annual evolution (Figure 9), it could be noticed that inbreeding variations in both “berrenda” breeds did not present the same rates. So, animals belonging to BC reference population that were born in 2018 showed a lower consanguinity rate than animals born in 2017, while BN animals that were born in 2018, appeared to be higher inbred than those from 2017.

The tendency to increase inbreeding (*F*) and differences between both “berrenda” cattle breeds might also be determined by the different phenotypic (and genotypic selection actions carried out for the eradication of recessive MC1R locus alleles in BN and the eradication of the 1;29 Robertsonian translocation (t1:29) [14,15] in both breeds and on the other hand, a consequence of the increase in the quality of pedigree information. It should be noted that if higher inbreeding values were estimated in more recent periods due to an improvement in pedigree information, this would lead to overestimate the annual increase in inbreeding rate (Figure 9).

Minimize inbreeding such as the exchanging of breeding animals among herds or the establishment of mating programs, are limited by different reasons: (i) diseases such as tuberculosis or blue tongue are limiting livestock movements among the main geographical areas where the “berrenda” cattle herds are located (Andalucía, Extremadura and Castilla-La Mancha); (ii) mindset of some livestock farmers, most of them also owners of bullfighting farms, that are really involved in breeding their own lineages and that are reluctant to share their breeding animals with other farmers; (iii) general problems linked to the implementation of reproductive programs based on artificial insemination, also linked to a lack of frozen semen doses (at this moment, only semen doses from 18 BN sires and from 27 BC sires are stored in germplasm banks).

In view of the problems for implementing rotational or systematic breeding schemes addressed to tackle inbreeding rates by maintaining family groups, ANABE has been providing breeders reports on coancestry rates between males and females at herd level. This plan is flexible and does not restrict the use of any of the sires, allowing breeders themselves to design and program the most appropriate matings, in order to minimize kinship between breeding animals at herd level thanks to a more diverse use of sires [35].

BN and BC effective population sizes were estimated for their respective reference populations, based on family size variance (*N_es_*) (BN = 147.05 and BC = 190.18); and also, they were estimated for the corresponding individual increasing of inbreeding for a similar contribution of founders (*N_ei_*) [52,53], being in a general way, low in both “berrenda” cattle breeds (9.93 ± 3.76 in BN and 11.58 ± 4.81 in BC).

The annual evolution of BN and BC *N_es_* values when considering their reference populations (Figure 10), have shown a very irregular trend. Wainwright et al. [80] suggested the use of *N_es_* and their contribution over 5 years to analyze the evolution of conservation activities.

Considering the imbalances affecting the information about different ancestral pathways provided by the pedigrees of the “berrenda” cattle breeds, metrics as *N_ec_* based on their respective coancestries according to Cervantes et al. [56] could be a good option for monitoring both populations and for detecting recent changes that could be affecting genetic variability [55]. This parameter varied from 92.28 ± 4.21 in BN to 169.92 ± 5.74 in BC (Table 4) and the values were about 10 times higher than *N_ei_* in both breeds, what could suggest that these populations were sub-structured (Wahlund’s effect).

### 3.4. Probability of Genes Origin and Ancestral Contributions

Founders’ background in the present populations of the “berrenda” cattle breeds is intended to be preserved. A small part of the genetic diversity might be lost because of the unbalanced contributions of founders and also to the effects of bottlenecks over time that populations could have experienced historically, such as those caused by pyramid selection schemes, also based on the use of artificial insemination [13,81]. An effective number of ancestors (*f_a_*) lower than the effective number of founders (*f_e_*), what would bring to light a decrease in genetic variation derived from bottlenecks increasing the susceptibility to diseases or by reducing its resilience to climate change [82].

Considering that BC population is much larger than BN population, the values obtained for the effective number of founders (*f_e_*) and for the effective number of ancestors (*f_a_*) were also higher (Table 5). So, in BN whole population, the calculated value for *f_e_* was 88 and for *f_a_* was 87. In the case of BC, the calculated values for *f_e_* and *f_a_* were 238 and 234, respectively. So, in both breeds, ratio *f_e_*/*f_a_* was estimated to be close to 1.00, what would suggest no differences between the contributions of ancestors and founders, no trace of bottlenecks or even, a scarce loss of information about founders’ contributions.

In addition, equivalent number of founder genomes (*f_g_*) and *f_g_*/*f_e_* ratio would not suggest any unbalanced contribution of founders that could lead to a loss in genetic diversity; these parameters were slightly higher in BN, what would point out a more intensive use of certain breeding animals in comparison to BC and that could also be determined by a higher average inbreeding or a lower number of individuals under study. On the other hand, *f_g_*/*f_a_* ratio was also close to 1, what would highlight no evidence of genetic drift effects on the loss of genetic variability in these breeds. Rural mechanization process has been blamed as responsible for the bottlenecks suffered by the majority of the European autochthonous cattle breeds. “Berrenda” cattle breeds were also affected by this process and but these two breeds showed a high genetic variability, that apparently had not been affected too much over the years; furthermore, apparently there had not been an excessive unbalanced use of breeding animals.

On the other hand, *f_e_*/*f_a_* coefficient values calculated in both “berrenda” cattle breeds, quite similar to those reported for different Spanish breeds, such as the Avileña Negra Ibérica (1.15) and Morucha (1.24) breeds [42] and Portuguese breeds, such as the Mertolenga (1.48) and Preta (1.28) breeds [15]; what revealed that none of the “berrenda” breeds have suffered remarkable effects of bottlenecks throughout generations. This situation is quite uncommon in endangered breeds, however, there are many other trans-boundary meat breeds such as the Charolais and Limousin breeds, that despite their large census size, they have suffered the effects of bottlenecks, as a consequence of the implementation of genetic selection programs, showing *f_e_*/*f_a_* coefficient values above 2.5 [13].

Some differences were observed in BN and BC between the contributions of founders to the current reference population and the contributions of founders to the total population (Table 5). The number of ancestors needed to explain 50% of genes pool in the whole population (*N_a50_*) was 50 in BN and 101 in BC. Similarly, in the reference population, only 36 BN ancestors (6.38% of the population) and 57 BC ancestors (7.01% of the population) were needed to justify 50% of the genetic variability. These results would point out a lower genetic base in BN and also that BN contribution to genetic diversity within the breed was lower than BC contribution. *N_a50_* values in BN reference population was lower than those from the seven Spanish meat breeds studied by Cañas-Álvarez [31] and *N_a50_* value in BC was very similar to the value observed in the Spanish Avileña Negra Ibérica breed (52) [31] and to the value observed in the Portuguese Mertolenga breed (56) [13].

Among the five main ancestors that contributed the most to BN and BC reference populations genetic variability, only one female was found in BC (Table 6). In BN, the most important animal was a founder male born in 1999, contributing 0.06% to the genetic variability of the population, with 264 descendants throughout the generations. In general, the five main ancestors in BN had a contribution of 0.67% to the genetic variability of the population, contributing 971 descendants to it. In the case of BC, a male born in 2003 contributed 315 descendants to the population, representing a genetic contribution of 0.02%.

Alderson’s Genetic Conservation Index (*GCI*) [4] was used as a parameter to guide conservation programs and to monitor genetic diversity among breeds, by pointing out the average number of founders per individual. In both cases, the calculated values for this parameter were quite low, getting an average *GCI* value of 2.80 for the whole BN population and 2.82 for the whole BC population (Table 5).

The whole genetic diversity loss (*GD* *–*GD*) varied between 0.09% and 0.01%. The 1-*GD* loss (derived from bottlenecks and genetic drift) in the reference population since the founder population, got a value of 0.4% in BN and 0.1% in BC, respectively. The 1-*GD* * loss that considers the unequal contribution of founders was equal to 0.5% in BN and equal to 0.2% in BC. The latter might be the main cause of loss of genetic diversity in the “berrenda” cattle breeds.

Regarding the number of herds in the “berrenda” cattle breeds, only 11 herds contributed to 50% of the current gene pool in the BN reference population and the effective number of herds contributing parents, grandparents and great-grandparents to the population (Table 7) was 36.97, 16.91 and 8.18, respectively. Obviously, BC doubled BN values for those parameters (66.40, 36.47 and 19.06, respectively) and 22 herds contributed to 50% of the current gene pool in the BC reference population. Ratio between herd number and effective number of founder herds (31.3% and 42.2% in BN and BC, respectively) was close to the established ratio for other breeds reared in the Dehesa ecosystem (40%) with a low migration rate from one herd to another [42].

All BN and BC herds were using purchased sires and none of them was considered as isolated or as nucleus herd according to Robertson’s [44] classification (Table 8). Multiplier herds that were providing sires to other herds were the ones contributing the most to the dissemination of the genetic progress, representing 56.8% in BN and 51.0% in BC, respectively. On the other side, commercial herds represented 43.2% in BN and 49.0% in BC, respectively, although most of them were also using sires born and raised in the same farm (commercial herds type A). The group made by commercial herds type B was quite large in numeric terms, especially in BC, but herds selling oxen might be included. Similarly, to commercial herds type A, they also appeared to be an additional reservoir of genetic variability, since they could potentially be considered as multiplier herds type A, taking part in the genetic business by spreading different genetic lines not employed yet.

Multiplier herds type B (herds selling bulls as sires but only using purchased bulls as sires) were not relevant in number. This situation might be determined by the fact that 15% of BN and 25% BC breeders were also rearing Bullfighting cattle (“Lidia” cattle) and management of cattle of this kind is usually based on the preservation of their own familiar lineages (“encastes”) [44]; bullfighting breeders’ mindset has been transferred and put into practice when planning and designing both the reproduction and mating programs at “berrenda” cattle farm level, which were identified as “commercial herds type A”. Restrictions on the movement of cattle among different geographical regions established by official health regulations and troubles for the implementation of Artificial Insemination in “berrenda” breeds, were also key factors that were limiting the exchange of breeding animals among herds and also complicating the implementation of strategies addressed to facilitate those exchanges.

The existing variability between breeds, linked to the possible differences among herds was analyzed according to Wright’s F statistics. So, considering the calculated mean values for F_ST_ (0.08 in BN and 0.075 in BC), apparently there would not be any subdivision into different subpopulations at herd level; related to F_IS_ (−0.044 in BN and −0.046 in BC), negative values could mean that the average inbreeding rate (*F*) within a herd could be lower than the average inbreeding rate among individuals, in both “berrenda” breeds, pointing out that average inbreeding rate within a subpopulation would not exceed coancestry between individuals and that matings between related animals would have been avoided [83]. According to Fernández et al. [84], maintaining a subdivided population would have the advantage of reducing the risk of extinction and decreasing the possible health or accidental risks that could cause its extinction if it were just a single group. In addition, the maximum long-term genetic diversity of a population could be achieved by subdividing it into groups as far apart as possible [85] but it would be necessary to control and minimize the effects of inbreeding inside herds and also, to maximize genetic contributions of ancestors.

Expectations regarding the development and conservation of “berrenda” breeds would require that consumers and institutions could acknowledge their values and their potential to face future challenges related to possible changes in consumers’ preferences, to sustainable economy of the territories and to climate change. Production demands could only be met if the implementation of breeding programs developed measures to prevent inbreeding.

According to our results, the strategy of minimum coancestrality within herds that is being implemented yet in the “berrenda” cattle breeds, apparently is successfully maintaining their genetic variability throughout generations. The relative isolation that is affecting some BN and BC herds due to the lack of exchanges of breeding animals among herds and to the geographical isolation established by animal health regulations may also be contributing to keep the original genetic variability within herds; but in order to tackle both loss of diversity trends and long-term increase in inbreeding that have been detected, it would be advisable to foster measures for the exchange of breeding animals among farms. Considering that probably, this could not be feasible in all herds, it would be necessary to support the implementation of complementary measures such as artificial insemination or an appropriate control of matings such as planning compensatory matings by minimizing coancestry, prolonging the reproductive use of the breeding animals but limiting the maximum offspring per couple that could lead to a significant reduction of inbreeding in the future [60].

The choice of sires and dams to provide germplasm for in vitro conservation, as well as the selection initiatives to eliminate animals carrying t 1:29 disorder or undesirable alleles of the MC1R gene should be carried out by applying optimal contribution selection approaches, in order to achieve these objectives without increasing individual inbreeding.

For all these and future actions, the current information provided by pedigrees of “berrenda” breeds would be sufficient and should be completed in the short term with the information provided by genomic tests that would allow monitoring genetic variability (such as ROH) and identifying genes linked to traits of economic interest, to disease resistance or to climate change.

## 4. Conclusions

The analysis of the population structure of the two berrenda cattle breeds reared in extensive farming systems (“Dehesa” eco-system) revealed that they are characterized by a lack of precocity, a low number of offspring but the reproductive longevity of dams.

According to our results, apparently pedigrees of these two breeds appeared to be not too deep, but they have been improved throughout the last seven years, what highlighted that the paternity monitoring established in their conservation programs was a remarkable success, thanks to the increasing number of animals with known genealogical information.

Although it should be noted their rather shallow pedigrees, the average values for inbreeding and relationship coefficient estimated for both “berrenda” cattle breeds would suggest a dangerous and increasing situation of loss in their genetic variability that could be even worse over time if no additional measures were implemented. In order to prevent future losses, highlighting that the situation in BN is even worse than in BC.

Currently, both the rate and the level of inbreeding in the “berrenda” breeds are lower than others’ cattle breeds that are reared in the “Dehesa” ecosystem; unfortunately, these trends would show that they would have been suffering some loss of genetic variability what would make necessary the implementation of certain measures addressed to control inbreeding.

Our results showed a high presence of founder genotypes that have been maintained in the current “berrenda” cattle populations at herd level. The strategy of minimum coancestrality within herds that has been implemented in the “berrenda” cattle breeds, apparently would be successfully maintaining their genetic variability.

The relative isolation that has been affecting some herds as a consequence of the limited exchanges of breeding animals among herds, the lack of implementation of assisted reproductive techniques, such as artificial insemination (IA) and also the geographical isolation established by animal health regulations, might also be contributing to the maintenance of the original genetic variability within herds; but in order to tackle both the loss of diversity trends and the long-term increase in inbreeding, it would be advisable to foster the exchange of breeding animals among farms.

In this work, we have analyzed for the first time the genealogy of the two Spanish autochthonous “berrenda” cattle breeds, whose analyses provided very useful information not only to the conservationists dealing with these breeds but also to ANABE breeders. It would be necessary the implementation of suitable institutional financial support that allowed ANABE not only to continue with the adequate management of pedigree information, but also to apply genomic methodologies to achieve the conservation objectives and address future selection actions in the “berrenda” cattle breeds.

## Figures and Tables

**Figure 1 animals-12-00249-f001:**
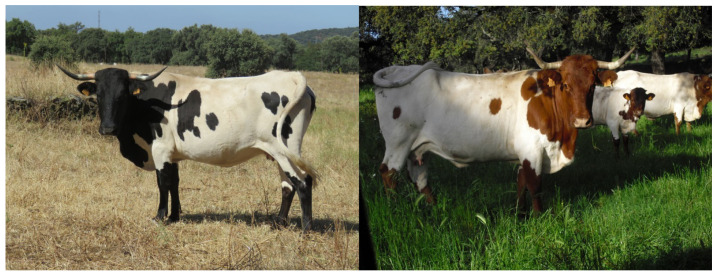
Cows of “Berrenda en Negro” and “Berrenda en Colorado” cattle breeds (rights owned by ANABE).

**Figure 2 animals-12-00249-f002:**
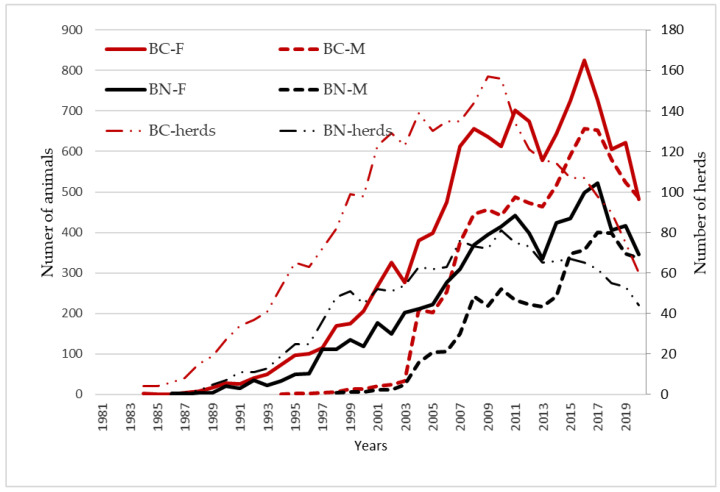
Number of registered males (M) and females (F) in the Berrenda en Negro (BN) and Berrenda en Colorado (BC) studbooks by birth date and number of herds.

**Figure 3 animals-12-00249-f003:**
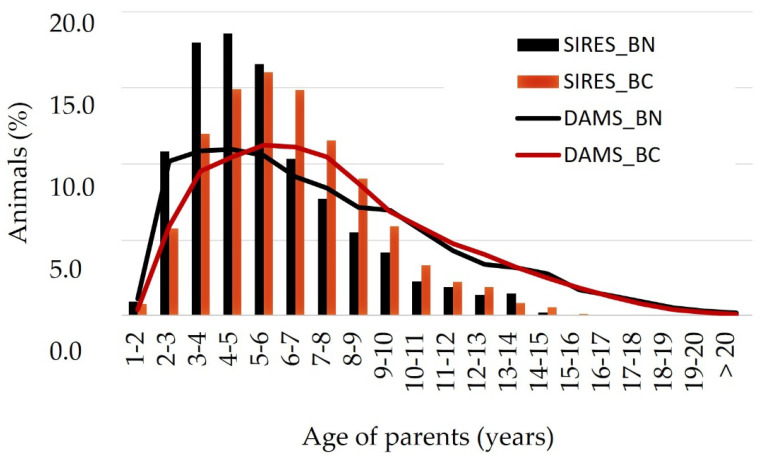
Age of the “Berrenda en Negro” (BN) and “Berrenda en Colorado” (BC) parents when their registered off-spring are born.

**Figure 4 animals-12-00249-f004:**
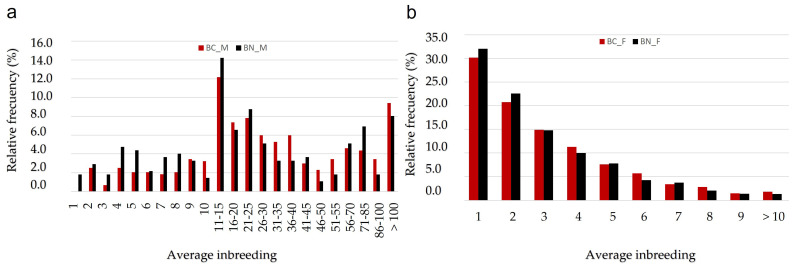
Relative frequency in the number of descendants per (**a**) sire and (**b**) dam in both “berrenda” breeds.

**Figure 5 animals-12-00249-f005:**
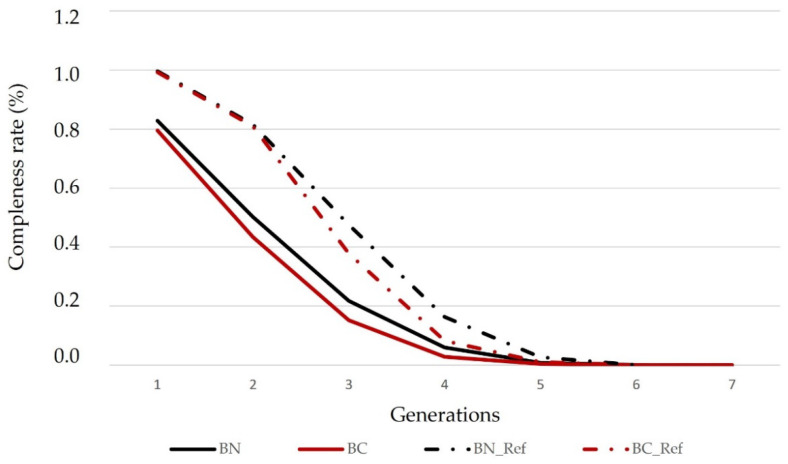
Average percentage of known ancestors per generation in BN and BC whole populations and in the reference populations of the “Berrenda en Negro” (BN) and “Berrenda en Colorado” (BC) cattle breeds (born animals between 2013 and 2020).

**Figure 6 animals-12-00249-f006:**
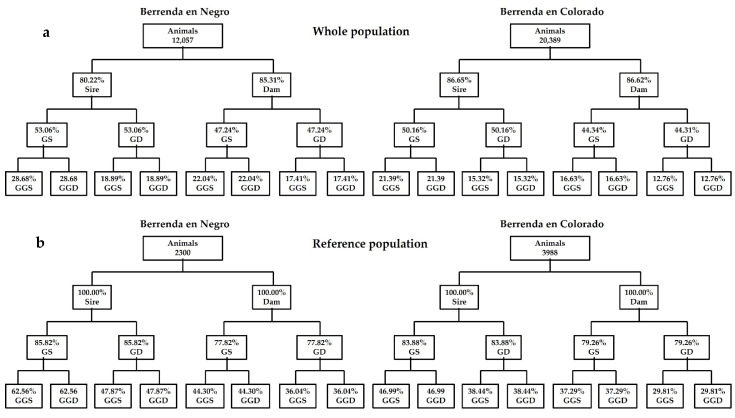
Average percentage of known ancestors per complete generations (Sire and Dam = parents; *GD* = grandam; GS = grandsire; GGD = great-grandam; GGS = great-grandsire) in the Berrenda en Negro and Berrenda en Colorado pedigrees; (**a**) whole populations and (**b**) reference populations.

**Figure 7 animals-12-00249-f007:**
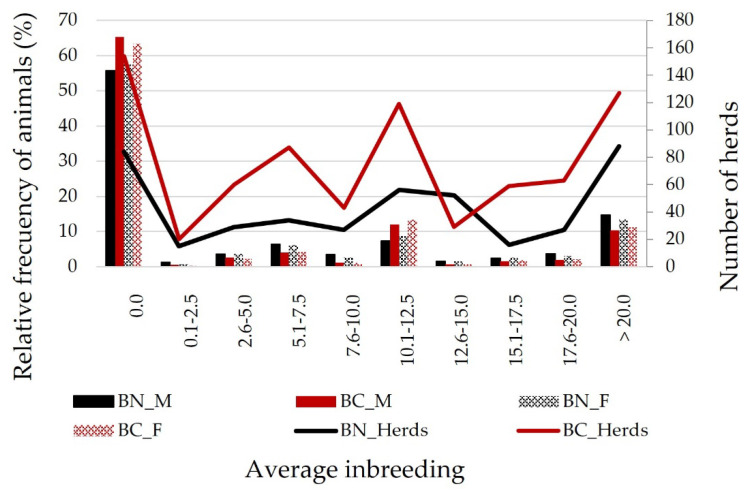
Average inbreeding rates (%) in herds, males (M) and females (F) from reference populations of “Berrenda en Negro” (BN) and “Berrenda en Colorado” (BC) cattle breeds.

**Figure 8 animals-12-00249-f008:**
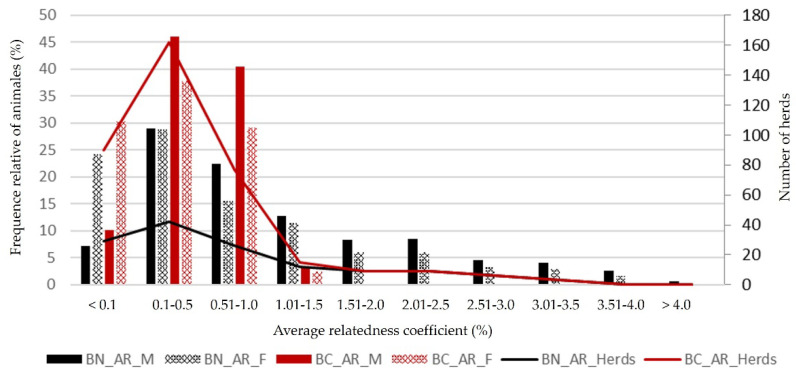
Average relationship (*AR*) coefficients inbreeding rates (%) in herds, males (M) and females (F) from reference populations of “Berrenda en Negro” (BN) and “Berrenda en Colorado” (BC) cattle breeds.

**Figure 9 animals-12-00249-f009:**
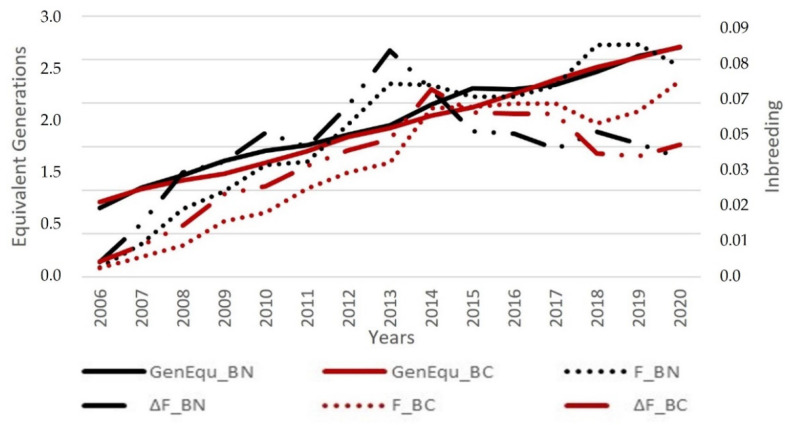
Average inbreeding values (*F*), average individual increase of inbreeding (*∆F*) and average equivalent generations (*GenEqu*) in the “Berrenda en Negro” (BN) and “Berrenda en Colorado” (BC) studbooks per year of birth of the individuals belonging to the reference populations.

**Figure 10 animals-12-00249-f010:**
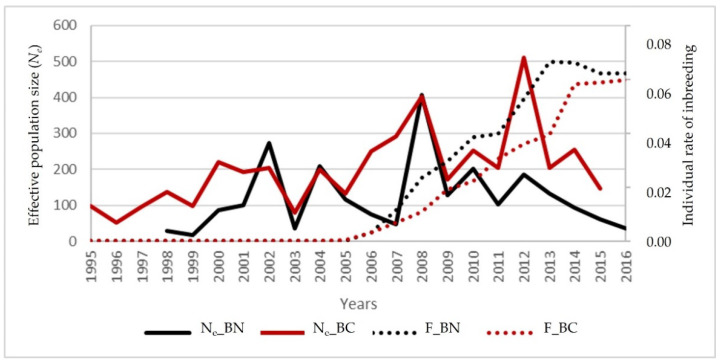
Evolution per year of birth of the average Individual rate of inbreeding (*F*) and the Effective population size (*N_e_*) in “Berrenda en Negro” (BN) and “Berrenda en Colorado” (BC) whole populations.

**Table 1 animals-12-00249-t001:** Generation interval (*L*) in terms of “years” and average age of breeding animals when their offspring is born in the “Berrenda en Negro” (BN) and “Berrenda en Colorado” (BC) breeds.

	Total Population						Reference Population					
	Generation Interval			Average Age of Breeding Animals			Generation Interval			Average Age of Breeding Animals		
	N	Mean ± SE	SD	N	Mean ± SE	SD	N	Mean ± SE	SD	N	Mean ± SE	SD
Sire-Son												
BN	187	4.92 ± 0.17	2.30	3902	5.54 ± 0.04	2.49	36	5.47 ± 0.45	2.68	514	5.92 ± 0.12	2.81
BC	264	5.58 ± 0.15	2.50	7810	6.04 ± 0.03	2.68	40	6.39 ± 0.47	2.99	1020	6.21 ± 0.09	2.73
Sire-Daughter												
BN	1912	5.40 ± 0.05	2.40	5770	5.70 ± 0.03	2.64	599	5.79 ± 0.49	2.96	1786	6.02 ± 0.13	2.99
BC	3083	5.72 ± 0.04	2.45	9857	6.02 ± 0.03	2.71	914	6.43 ± 0.46	2.91	2968	6.36 ± 0.09	2.92
Dam-Son												
BN	187	6.70 ± 0.26	3.60	4203	7.55 ± 0.06	4.03	36	6.67 ± 0.73	4.38	514	7.81 ± 0.18	4.18
BC	264	7.35 ± 0.23	3.72	7810	7.55 ± 0.04	3.87	40	7.68 ± 0.60	3.78	1020	7.89 ± 0.12	4.02
Sire-Daughter												
BN	1912	7.09 ± 0.09	3.96	6083	7.53 ± 0.05	4.09	599	7.27 ± 0.71	4.24	1786	7.72 ± 0.19	4.31
BC	3082	7.33 ± 0.07	3.91	9852	7.60 ± 0.04	3.98	914	7.40 ± 0.60	3.77	2968	7.77 ± 0.13	4.01
Total												
BN	4198	6.20 ± 0.05	3.36	19,958	6.62 ± 0.03	3.56	1270	6.50 ± 0.10	3.73	4600	6.87 ± 0.06	3.78
BC	6693	6.52 ± 0.04	3.36	35,329	6.80 ± 0.02	3.46	1908	6.92 ± 0.08	3.40	7976	7.06 ± 0.04	3.57

**Table 2 animals-12-00249-t002:** Average inbreeding coefficient (*F*), percentage of inbred individuals (*Inb*), average F from inbred animals (*FInb*), average relationship coefficient (*AR*) and Effective population size * (*N_e_*) per generation in the populations of “berrenda” cattle breeds.

	Berrenda en Negro						Berrenda en Colorado					
Generation	*n*	*F* (%)	*Inb* (%)	*FInb* (%)	*AR* (%)	*N_e_*	*n*	*F* (%)	*Inb* (%)	*FInb* (%)	*AR* (%)	*N_e_*
0	2386	0			0.18		4235	0			0.02	
1	5250	2.25	8.59	26.18	0.80	22.2	10,517	1.98	8.00	24.79	0.33	25.2
2	3624	9.20	60.43	15.22	1.38	7.0	6426	6.72	47.82	14.06	0.53	10.3
3	774	11.88	84.24	14.11	2.34	16.9	718	8.57	63.79	13.43	0.64	25.2
4	23	12.10	100	12.10	2.45	202.9	1	19.53	100.00	19.53	0.58	4.1

*n* = number of individuals. * *N_e_* has been estimated from the individual increases of inbreeding, considering an equal contribution of founders, according to Gutiérrez et al. [39,40].

**Table 3 animals-12-00249-t003:** Average number of traced generations, increase of inbreeding (*ΔF*) and effective population size (*N_e_*) in the “Berrenda en Negro” and “Berrenda en Colorado” breeds reference populations, according to the generation type.

	Berrenda en Negro			Berrenda en Colorado		
	Maximum ^a^	Complete ^b^	Equivalent ^c^	Maximum ^a^	Complete ^b^	Equivalent ^c^
Average number of generations	2.15	1.24	1.62	1.97	1.25	1.54
Inbreeding rate (%) per generation (*ΔF*)	2.47	4.59	4.02	1.75	3.53	3.01
Effective Population Size (*N_e_*)	20.22	10.90	12.44	28.62	14.18	16.62

^a^ Number of generations between an animal and its closest ancestor. ^b^ The further generation which all ancestors are known. ^c^ Addition of the term ${\left(1/2\right)}^{n}$ from all known ancestors, where *n* equals to the number of generations separating one animal from every known ancestor.

**Table 4 animals-12-00249-t004:** Inbreeding rate per equivalent generation (*∆F*), Inbreeding after 10 (*F_10_*) and 50 (*F_50_*) years, Effective population size based on individual rate of inbreeding (*N_ei_*), Effective population size based on the increasing of the individual coancestry rate (*N_ec_*) and number of equivalent subpopulations in the reference population of Berrenda en Negro (BN) and Berrenda en Colorado (BC) cattle breeds.

	N	*ΔF*	*F_10_*	*F_50_*	*N_ei_*	*N_ec_*	Equivalent Subpopulations (*N_ec_/N_ei_*)
BN	1716	3.01	6.15	33.7	9.93 ± 3.76	92.28 ± 4.21	9.29 ± 3.54
BC	2968	4.02	4.34	18.78	11.58 ± 4.81	169.92 ± 5.74	14.67 ± 6.11

*N_ei_*: Effective size by increase of individual inbreeding. *N_ec_*: Effective size by increase of coancestrality.

**Table 5 animals-12-00249-t005:** Parameters that characterize the probability of genes origin in the Berrenda en Negro (BN) and Berrenda en Colorado (BC) breeds whole populations and in their reference populations.

	BN		BC	
Parameters ^1^	Total	Reference Population	Total	Reference Population
Total number of animals	12,057	2300	20,389	3988
Total number of animals with both known parents	9671		17,662	
Total number of founders	2386	732	2727	1240
Number of ancestors	1657	726	2741	1242
Average inbreeding coefficient	4.53	7.00	3.44	5.70
Average relationship	0.95		0.40	
Effective number of founders	88	58	238	140
Effective number of ancestors	87	57	234	133
Number of Founder Herds		75		135
Effective number of founder herds		23.9		60.9
Equivalent number of founder genomes	104.77		251.12	
*f_a_*/*f_e_*	1.01	1.01	1.01	1.05
*f_g_*/*f_e_*	1.19		1.05	
*f_g_*/*f_a_*	1.20		1.09	
Genetic Diversity	0.9952		0.9983	
*GD* Loss caused by bottle necks or genetic drift since founders level	−0.0013		−0.0003	
Number of ancestors explaining 50% of genetic variability	50	36	101	57
Genetic conservation Index	2.80		2.82	

^1^*f_a_* = Effective number of ancestors; *f_e_* = Effective number of founders; *f_g_* = Equivalent number of founder genomes.

**Table 6 animals-12-00249-t006:** Description of the five main ancestors * that have contributed the most to the whole population and to the reference populations of the “berrenda” cattle breeds.

Berrenda en Negro						Berrenda en Colorado				
Sex		Herd of Origin	Year of Birth	Genetic Contribution (%)	No. of Descendants	Sex	Herd of Origin	Year of Birth	Genetic Contribution (%)	No. of Descendants
Whole Population										
1	M	52	1999	0.06	264	M	84	2003	0.02	315
2	M	84	2003	0.11	276	H	162	2005	0.04	6
3	M	35	2001	0.14	54	M	2	2003	0.05	219
4	M	19	2003	0.17	287	M	72	2003	0.07	198
5	M	151	2003	0.19	90	M	8	2001	0.08	175
Reference Population										
1	M	84	2003	0.07	276	M	84	2003	0.04	315
2	M	52	1999	0.13	264	H	162	2005	0.07	6
3	M	35	2001	0.17	54	M	72	2003	0.08	198
4	M	151	2003	0.21	90	M	112	2006	0.1	265
5	H	52	2003	0.24	5	M	48	2006	0.12	98

* The five main ancestors and their genetic marginal contribution identified according to Boichard et al. [59].

**Table 7 animals-12-00249-t007:** Number of herds supplying ancestors in every generation (1 = parents, 2 = grandparents…) to the “berrenda” cattle breeds population.

Generation	Berrenda en Negro		Berrenda en Colorado	
	Current	Effective	Current	Effective
1	99	36.97	166	66.40
2	55	16.91	88	36.47
3	27	8.18	42	19.06
4	14	6.24	11	6.35
5	3	2.81	2	1.19

**Table 8 animals-12-00249-t008:** Number of herds in both “berrenda” cattle breeds classified per type of breeding animals’ exchanges and identified as nucleus, multiplier, commercial or isolated herds *.

Herd Types	Berenda en Negro					Berrenda en Colorado				
	UPB	UOB	SB	NH	PPB%	UPB	UOB	SB	NH	PPB%
Nucleus herd	No	Yes	Yes	0	0	No	Yes	Yes	0	0
Multiplier herd type A	Yes	Yes	Yes	62	35.37	Yes	Yes	Yes	116	35.63
Multiplier herd type B	Yes	No	Yes	5	100	Yes	No	Yes	4	100
Commercial herd type A	Yes	Yes	No	30	32.75	Tes	Yes	No	44	29.13
Commercial herd type B	Yes	No	No	22	100	Yes	No	No	46	100
Isolated herd	No	Yes	No	0	0	No	Yes	No	0	0.00
Total				119					210	

* Herd types are defined and classified according to type of breeding animals’ exchanges by Robertson [57]. UPB: Herds using purchased sires; UOB: Herds using their own sires; SB: Herds selling bulls used as sires; NH: Number of herds per herd type; PPB: Percentage of purchased sires.

## Data Availability

This is not applicable, as the data are not in any data repository with public access. However, if an editorial committee needs access, we will happily provide them with it; please use this email: pa1rosee@uco.es.

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
