# Peer review of "Analyses of Genetic Diversity in the Endangered “Berrenda” Spanish Cattle Breeds Using Pedigree Data"

_animals, 2022, doi:10.3390/ani12030249_

Round 1

Reviewer 1 Report

Dear Authors!

The issues of endangered local breeds conservation are important at the moment. The complexity of this process is associated with the low competitiveness of such breeds in comparison with global  specialized breeds. Restrictions on the intensity of breeding in existing populations of local breeds are imposed by the need to maintain genetic diversity on the one hand and the small number of animals in these populations, on the other hand.

In general, your research is of some interest from the point of view of a detailed description of the situation in the endangered Berrenda breeds. A number of the problems you are covering are relevant for many local populations. However, I would like to note two important general points:

- more effective methods have been developed to evaluate some of the parameters mentioned in your work, but they are based on genomic data; Why don't you mention these studies?

- in my opinion, the addition of the prospects for the development of the issue you are covering, the prospects for the development of the studied livestock breeds, your opinion on the expediency of preserving and maintaining these populations, would significantly increase your work value.

In addition, there are a number of questions and comments:

Lines 84-85         it is said about the preservation of breeds and the maintenance of their genetic diversity

                              «…conservation of the populations and the conservation of their genetic variability»

                              However

Lines 89-90        «…the improvement of the female breeding 89 skills from the calves’ growing rates and from the estimation of their Genetic Value (GV) 90 by BLUP. »

                              this requires your comment

lines 128-136     it is desirable to rewrite; since the analysis of pedigrees is, although simpler, but at the same time less accurate method. It may be worth noting that the use of genomic data makes it possible to more accurately assess the parameters: population structure, effective population size, degree of kinship; however, it requires significant financial costs.

Line 145              it is advisable to rewrite the goal. «The aim of this work was to study current genetic variability, ... based on deeply analysis of pedigrees…»

Lines 247-297    it is advisable to move text to the Introduction section

Lines 302-317    this description looks unnecessarily detailed and adds an unscientific sound to the work

Lines 329-330    is it worth talking about selection among females? Most likely, it is missing. This explains the higher age of females

and 344-345      

In my opinion, there is no need to cite Table 1 and Fig.3 and 4 at the same time. This is unnecessary, especially since the text contains a detailed description

Line 470-473      Fig.6 is redundant, it is advisable to remove it

Line 452 – 482   the text relating to the description of the completeness of pedigrees is unnecessarily detailed. It needs to be reduced. It makes it difficult to read the article

Lines 635-644    It is not quite correct to calculate the level of inbreeding, including data on animals without a reliable pedigree. It is more logical to form a data set of individuals with complete pedigrees, providing the number of excluded data.

Line 648              «… to make a huge effort in improving pedigree recordings… » gives an unscientific sound, replace on « emphasizes the importance of reliable data on kinship»

Line 855              what specific measures do the authors propose to apply to improve inbreeding control?

Line 857-861      does this mean that the results obtained cannot be considered reliable?

If yes, then it is desirable to remove this conclusion or completely reformulate it.

I would like to understand the authors' opinion on the importance and prospects of work on the preservation of Berrenda breeds. To what extent, in the authors' opinion, the use of genomic methods could contribute to the creation of a more objective and clear picture of the decline in genetic diversity in the studied breeds. The analysis used in this work is based on traditional, but somewhat outdated methods, and therefore it seems logical to add a description of the results of comparative studies based on "old" methods and more modern ones (for example, calculation of inbreeding based on pedigree and genomic inbreeding), reducing the descriptive part of some subsections. This, in my opinion, would significantly increase the work’s value.

Good Luck!

Author Response

RESPONSE Reviewer#1

Review Report Form

Open Review

(x) I would not like to sign my review report

( ) I would like to sign my review report

English language and style

( ) Extensive editing of English language and style required

( ) Moderate English changes required

( ) English language and style are fine/minor spell check required

(x) I don't feel qualified to judge about the English language and style

Yes

Can be improved

Must be improved

Not applicable

Does the introduction provide sufficient background and include all relevant references?

( )

( )

(x)

( )

Is the research design appropriate?

( )

( )

(x)

( )

Are the methods adequately described?

( )

(x)

( )

( )

Are the results clearly presented?

( )

( )

(x)

( )

Are the conclusions supported by the results?

( )

(x)

( )

( )

Comments and Suggestions for Authors

Dear Authors!

The issues of endangered local breeds conservation are important at the moment. The complexity of this process is associated with the low competitiveness of such breeds in comparison with global specialized breeds. Restrictions on the intensity of breeding in existing populations of local breeds are imposed by the need to maintain genetic diversity on the one hand and the small number of animals in these populations, on the other hand.

In general, your research is of some interest from the point of view of a detailed description of the situation in the endangered Berrenda breeds. A number of the problems you are covering are relevant for many local populations. However, I would like to note two important general points:

1/ More effective methods have been developed to evaluate some of the parameters mentioned in your work, but they are based on genomic data; Why don't you mention these studies?

The reviewer is right, genomic methods have proven to be more accurate than those based on pedigree and also they allow detecting ancestral changes, but considering that genomic analyses have additional financial costs, they have not been developed yet in the "berrenda" breeds. This study has been developed, based on the information collected for the management of the studbooks of the “berrenda” breeds, according to the breeding programs carried out by ANABE. Currently, pedigree records are still being used to analyze the genetic variability of endangered breeds.

According to reviewer's advice, in the new version of the manuscript (Introduction, Discussion and Conclusions) we have corrected this issue by highlighting the usefulness and convenience of carrying out genomic analyses as follows:

1/ Introduction:

“Certainly for these purposes, information provided by modern genomic methods is quite better than information provided by pedigree analyses [30-32].  Genomic data would allow for a more accurate assessment addressed to determine different parameters about the structure of the population, such as the Effective population size or the Inbreeding rate [26,27,33], with the added advantage that when using genomic parameters we would be describing the realized genetic variation and not the expected one. Genomic tools are also quite effective for the conservation of diversity in livestock breeds when they are applied to the study of the genomic linkage disequilibrium due to genetic (selection, mutation, genetic drift, non-random mating…) and non-genetic forces [34]. Genomic analyses demand high financial cost, so the usage of genomic data is not so common in small livestock breeds and analyses based on pedigree data are still the preferential option for monitoring livestock conservation programs, supported by a combination of traditional [35-37] and genomic selection strategies [26,38].”

2/ Results and discussion:

“But it is necessary to be careful in the control of the mating strategies; on the contrary, there would be a high risk of mating between animals belonging to the same lineage, what would lead to an increase of inbreeding [70]. High L values would lead to lower genetic progress and lower genetic gains (in terms of selection of desirable traits) [71]. Genomic selection allows carrying out selective actions addressed to improve some traits of economic interest on the basis of small reference population sizes and moderately long generation intervals, without finding significant differences in the expression of Bulmer’s effect between genomic selection and other traditional selection approaches based on BLUP methodology [72].”

“For all these and future actions, the current information provided by pedigrees of "berrenda" breeds would be sufficient and should be completed in the short term with the information provided by genomic tests that would allow monitoring genetic variability (such as ROH) and identifying genes linked to traits of economic interest, to disease resistance or to climate change.”

3/ Conclusions:

“In this work, we have analyzed for the first time the genealogy of the two Spanish autochthonous “berrenda” cattle breeds, whose analyses provided very useful information not only to the conservationists dealing with these breeds but also to ANABE breeders. It would be necessary the implementation of suitable institutional financial support that allowed ANABE not only to continue with the adequate management of pedigree information, but also to apply genomic methodologies to achieve the conservation objectives and address future selection actions in the "berrenda" cattle breeds.”

2/ In my opinion, the addition of the prospects for the development of the issue you are covering, the prospects for the development of the studied livestock breeds, your opinion on the expediency of preserving and maintaining these populations, would significantly increase your work value.

Following reviewer’s suggestions, we have clarified the aims of our work in the “Introduction” section as follows:

“As a contribution for the development of the conservation programs in both “berrenda” cattle breeds, this work aims to study the current genetic variability and the structure of BN and BC populations based on thorough analyses of their corresponding pedigrees, by quantifying the changes that happened over the time and comparing them to other different autochthonous cattle breeds that have also been traditionally reared in the “Dehesa” ecosystems, following a quite similar farming system.

To be more precise, the objective of this work are : i) characterizing the structure of BN and BC populations by ages and sex and type of herd; ii) identifying the genetic variability level and the threat of inbreeding in both populations; iii) identifying the genes origin in both populations. It is also intended to know the possible loss of genetic.

This information will be quite useful to establish the most suitable strategies to be implemented in the conservation and selection programs of these valuable genetic resources in risk of extinction.”

Also, we have discussed about the prospects for the development of the “berrenda” breeds and the best way to implement their breeding programs. As a consequence, we have made some changes at the end of the “Results and discussion” section, included in the new version of the manuscript:

“Expectations regarding the development and conservation of "berrenda" breeds would require that consumers and institutions could acknowledge their values and their potential to face future challenges related to possible changes in consumers’ preferences, to sustainable economy of the territories and to climate change. Production demands could only be met if the implementation of breeding programs developed measures to prevent inbreeding.

According to our results, the strategy of minimum coancestry within herds that is being implemented yet in the “berrenda” cattle breeds, apparently is successfully maintaining their genetic variability throughout generations. The relative isolation that is affecting some BN and BC herds due to the lack of exchanges of breeding animals among herds and to the geographical isolation established by animal health regulations may also be contributing to keep the original genetic variability within herds; but in order to tackle both loss of diversity trends and long-term increase in inbreeding that have been detected, it would be advisable to foster measures for the exchange of breeding animals among farms. Considering that probably, this could not be feasible in all herds, it would be necessary to support the implementation of complementary measures such as artificial insemination or an appropriate control of matings such as planning compensatory matings by minimizing coancestry, prolonging the reproductive use of the breeding animals but limiting the maximum offspring per couple that could lead to a significant reduction of inbreeding in the future [61].

The choice of sires and dams to provide germplasm for in vitro conservation, as well as the selection initiatives to eliminate animals carrying t 1:29 disorder or undesirable alleles of the MC1R gene should be carried out by applying optimal contribution selection approaches, in order to achieve these objectives without increasing individual inbreeding.”

3/ Lines 84-85: It is said about the preservation of breeds and the maintenance of their genetic diversity:   «…conservation of the populations and the conservation of their genetic variability»

However, in Lines 89-90:  «…the improvement of the female breeding skills from the calves’ growing rates and from the estimation of their Genetic Value (GV) by BLUP. »

This requires your comment.

We are really sorry about this misunderstanding.

GVs have only been calculated in animals belonging to herds whose owners have voluntarily decided to carry out performance controls both in farms and testing centres. But controls are not compulsory to be performed and this issue is not the aim in any of the breeding programs. To be more precise, EBVs have been calculated in some animals but selection measures are not being implemented in order to increase meat production at breed level. Next year, it is intended to start performing artificial insemination in both breeds, and according to that, it would be fundamental to consider the implementation of measures addressed to preserve Genetic Variability in these two breeds.

In order to clarify all this, we have rewritten the next paragraph in the “Introduction” section as follows:

“Moreover, ANABE is planning to implement a program in coming years addressed to improve females rearing skills by analyzing calves’ growing rates and by calculating their estimated breeding value (EBV) using BLUP, the genetic estimation method according to the algorithm by Meuwissen et al. [24]. Semen collection and freezing (for in vitro conservation purposes) are being carried out from those bulls with the highest genetic variability, no t1:29 carriers and in some cases, with their corresponding EBVs. Also, ANABE is planning to start an artificial insemination (AI) program in 2022 addressed to preserve Genetic Variability.”

4/ Lines 128-136: It is desirable to rewrite; since the analysis of pedigrees is, although simpler, but at the same time less accurate method. It may be worth noting that the use of genomic data makes it possible to more accurately assess the parameters: population structure, effective population size, degree of kinship; however, it requires significant financial costs.

These corrections have already been done and this issue has been commented previously.

5/ Line 145: It is advisable to rewrite the goal. «The aim of this work was to study current genetic variability ... based on deeply analysis of pedigrees…»

We have rewritten the text as follows:

“As a contribution for the development of the conservation programs in both “berrenda” cattle breeds, this work aims to study the current genetic variability and the structure of BN and BC populations based on thorough analyses of their corresponding pedigrees, by quantifying the changes that happened over the time and comparing them to other different autochthonous cattle breeds that have also been traditionally reared in the “Dehesa” ecosystems, following a quite similar farming system.

To be more precise, the objective of this work are : i) characterizing the structure of BN and BC populations by ages and sex and type of herd; ii) identifying the genetic variability level and the threat of inbreeding in both populations; iii) identifying the genes origin in both populations. It is also intended to know the possible loss of genetic.

This information will be quite useful to establish the most suitable strategies to be implemented in the conservation and selection programs of these valuable genetic resources in risk of extinction.”

6/ Lines 247-297: It is advisable to move text to the Introduction section.

This part has been moved to the “Introduction” section and it has been modified as previously explained.

7/ Lines 302-317: This description looks unnecessarily detailed and adds an unscientific sound to the work.

According to reviewer’s indications, we have modified the mentioned paragraphs in order to reduce them, as follows:

“From that moment, falls in the number of registrations, could be associated on the one side, to the strong dependence of these breeds on EU common agricultural policy subsidies for endangered autochthonous breeds [19,35,66,67] and on the other side, to the effects of the amendments in the Spanish regulation on the registration of new founder males in the studbooks from 2013 onward [8].”

8/ Lines 329-330: Is it worth talking about selection among females? Most likely, it is missing. This explains the higher age of females.

In line with reviewer’s comment, we have rewritten these lines as follows:

 “The highest age of females was expected, because selection strategies have largely relied on sires”.

9/ And 344-345:  In my opinion, there is no need to cite Table 1 and Fig.3 and 4 at the same time. This is unnecessary, especially since the text contains a detailed description.

Pursuant to reviewer’s opinion, Table 1 has been removed.

10/ Line 470-473: Fig.6 is redundant, it is advisable to remove it

With due respect, we consider that Figure 6 is complementary to Figure 5, providing additional information about the four different pathways (Sire-Son, Sire-Daughter, Dam-Son,Dam-Daughter).

11/ Line 452 – 482: the text relating to the description of the completeness of pedigrees is unnecessarily detailed. It needs to be reduced. It makes it difficult to read the article.

This paragraph has been removed, according with the indication of reviewer.

12/ Lines 635-644: It is not quite correct to calculate the level of inbreeding, including data on animals without a reliable pedigree. It is more logical to form a data set of individuals with complete pedigrees, providing the number of excluded data.

According to the previous suggestion, the above mentioned paragraph has been removed. Certainly, the reference population is made up by a highly reliable data set. Unfortunately, these data are not quite deep in order to consider the estimated effective population sizes based on ∆F. However, this limitation has been exposed in the manuscript, in order to

1/ To discuss the results:

“The tendency to increase inbreeding (F) and differences entre las both "berrenda" cattle breeds might also be determined by the different phenotypic (and genotypic selection actions carried out for the eradication of recessive MC1R locus alleles in BN and the eradication of the 1;29 Robertsonian translocation (t1:29) [14,21] in both breeds, and on the other hand, a consequence of the increase in the quality of pedigree information. It should be noted that if higher inbreeding values were estimated in more recent periods due to an improvement in pedigree information, this would lead to overestimate the annual increase in inbreeding rate (Figure 9).”

2/ Express a need:

“In this work, we have analyzed for the first time the genealogy of the two Spanish autochthonous “berrenda” cattle breeds, whose analyses provided very useful information not only to the conservationists dealing with these breeds but also to ANABE breeders. It would be necessary the implementation of suitable institutional financial support that allowed ANABE not only to continue with the adequate management of pedigree information, but also to apply genomic methodologies to achieve the conservation objectives and address future selection actions in the "berrenda" cattle breeds.”

13/ Line 648: «… to make a huge effort in improving pedigree recordings… » gives an unscientific sound, replace on « emphasizes the importance of reliable data on kinship»

That line has been removed in order to reduce and better organize the “Results and discussion” section, following reviewer’s piece of advice.

14/ Line 855: What specific measures do the authors propose to apply to improve inbreeding control?

The specific measures to improve inbreeding control have been mentioned at the end of the “Results and discussion” section, as follows:

“According to our results, the strategy of minimum coancestrality within herds that is being implemented yet in the “berrenda” cattle breeds, apparently is successfully maintaining their genetic variability throughout generations. The relative isolation that is affecting some BN and BC herds due to the lack of exchanges of breeding animals among herds and to the geographical isolation established by animal health regulations may also be contributing to keep the original genetic variability within herds; but in order to tackle both loss of diversity trends and long-term increase in inbreeding that have been detected, it would be advisable to foster measures for the exchange of breeding animals among farms. Considering that probably, this could not be feasible in all herds, it would be necessary to support the implementation of complementary measures such as artificial insemination or an appropriate control of matings such as planning compensatory matings by minimizing coancestrality, prolonging the reproductive use of the breeding animals but limiting the maximum offspring per couple that could lead to a significant reduction of inbreeding in the future [61].

The choice of sires and dams to provide germplasm for in vitro conservation, as well as the selection initiatives to eliminate animals carrying t 1:29 disorder or undesirable alleles of the MC1R gene should be carried out by applying optimal contribution selection approaches, in order to achieve these objectives without increasing individual inbreeding.

For all these and future actions, the current information provided by pedigrees of "berrenda" breeds would be sufficient and should be completed in the short term with the information provided by genomic tests that would allow monitoring genetic variability (such as ROH) and identifying genes linked to traits of economic interest, to disease resistance or to climate change.”

15/ Line 857-861: Does this mean that the results obtained cannot be considered reliable?

If yes, then it is desirable to remove this conclusion or completely reformulate it.

Answer: We have removed from the “Conclusions” section, the comments related to the comparisons with other breeds. According to that we have included the next paragraph:

“Our results showed a high presence of founder genotypes that have been maintained in the current “berrenda” cattle populations at herd level. The strategy of minimum coancestrality within herds that has been implemented in the “berrenda” cattle breeds, apparently would be successfully maintaining their genetic variability.”

16/ I would like to understand the authors' opinion on the importance and prospects of work on the preservation of Berrenda breeds. To what extent, in the authors' opinion, the use of genomic methods could contribute to the creation of a more objective and clear picture of the decline in genetic diversity in the studied breeds. The analysis used in this work is based on traditional, but somewhat outdated methods, and therefore it seems logical to add a description of the results of comparative studies based on "old" methods and more modern ones (for example, calculation of inbreeding based on pedigree and genomic inbreeding), reducing the descriptive part of some subsections. This, in my opinion, would significantly increase the work’s value.

In accordance with previous interviewer’s comments:

1/ In first place, we have highlighted in the “Introduction” section, both the importance and perspectives regarding the conservation of “berrenda” breeds; to be more precise:

“In summary, these are two autochthonous cattle breeds of high conservation interest, due to multiple reasons: historical, religious and cultural, linked to local traditional events that are important for the socioeconomic development of the regions where they are reared, mainly linked to tourism activities. They also contribute to the sustainable use and maintenance of the “Dehesa” unique ecosystems, increasing their environmental and economic value by producing meats of differentiated quality, linked to specific type of animals that are reared to be slaughtered (oxen). In addition, as ancestral breeds, they are also of special interest from the genetic point of view. The prospects for the development and conservation of these breeds rely on the ability of consumers and institutions to recognize these values and on the potential of these breeds to face future challenges related to possible changes in consumers’ preferences, to sustainable economy of the territories and to adaptation to climate changes. Market demands could only be met if the implementation of breeding programs developed measures addressed to the prevention of inbreeding.”

2/ Secondly, throughout the manuscript (Introduction, Discussion and Conclusions sections) we have proposed different conservation measures that could be implemented. On the other side, we have remarked our interest in using modern genomic methods which would complement and reinforce the security of the information and that could be also useful for the implementation of some other measures addressed to the conservation and selection of the breeds.

3/ Finally, we have reduced the size of different subsections in the manuscript, in order to remove all the information that could be considered as unnecessary or too descriptive.

Reviewer 2 Report

Journal: Animals (ISSN 2076-2615)

Manuscript ID: animals-1510819

Type: Article

Title: Analyses of genetic diversity in the endangered “Berrenda” Spanish cattle breeds using pedigree data

Major comments:

The major concern in the manuscript is pedigree completeness. From lines 469 to 474.

Whole population vs reference population.

All the parameters are influenced by the definition of founders in the genealogical information based on Pedigree.

When genomic data is used, the founders depend on the number of markers used in the analysis.

Minor comments:

From lines 48 to 52. Please add a reference.

From lines 54 to 55. Figure 1. Please insert photos where the morphology of the breeds could be appreciated.

From lines 63 to 67. Please add a reference. Please give more support to this idea. How do authors understand that both breeds maintain the ecosystem equilibrium?

From lines 89 to 94. Please add a reference. Please give more details to support this idea.

From lines 91 to 92. Authors mention, “Semen collection and freezing for in vitro conservation purposes are being carried out from bulls with the highest GV”. For genetic conservation, this strategy could bias the variability represented in the semen bank. Did the geneticist take into account the Bulmer effects?  What happens with the other unmeasured traits in the breeds?

From lines 102 to 110. Please add a reference. Please give more details to support this idea. Both paragraphs must be re-written.

From lines 142 to 152. Please clarify the objectives of the study. Look in some way something like considerations or conclusions.

From lines 155 to 158. This sentence looks like the one in lines 111 to 113. Please avoid repeating information.

From lines 171 to 174. Please add a reference.

From lines 176 to 179. Please add a reference.

Additional questions: did the authors consider that marginal genetic contribution estimated by PEDIG software could be relevant in this study? If yes, please incorporate it. If it is not, please widely argue why not.

From line 259 to 261. Here authors mention something about genomic data. When pedigree is not available or the completeness is not suitable for pedigree analysis. Genomics data could be used instead of pedigree. With the advantage that when genomics is used we are describing the realized genetic variation and not the expected. These arguments must be clarified when genomics data are mentioned in the manuscript.

Line 298. Here authors use reproductive structure. I am not sure but I can assume that the authors refer to population structure?

I suggest that all the results and discussions be re-written to clarify the presentation of the results.  Please try to argue properly all the concepts.

Author Response

RESPONSE Reviewer#2

Review Report Form

Open Review

(x) I would not like to sign my review report

( ) I would like to sign my review report

English language and style

( ) Extensive editing of English language and style required

(x) Moderate English changes required

( ) English language and style are fine/minor spell check required

( ) I don't feel qualified to judge about the English language and style

Yes

Can be improved

Must be improved

Not applicable

Does the introduction provide sufficient background and include all relevant references?

( )

( )

(x)

( )

Is the research design appropriate?

( )

(x)

( )

( )

Are the methods adequately described?

(x)

( )

( )

( )

Are the results clearly presented?

( )

(x)

( )

( )

Are the conclusions supported by the results?

( )

(x)

( )

( )

Comments and Suggestions for Authors

Journal: Animals (ISSN 2076-2615)

Manuscript ID: animals-1510819

Type: Article

Title: Analyses of genetic diversity in the endangered “Berrenda” Spanish cattle breeds using pedigree data

Major comments:

The major concern in the manuscript is pedigree completeness. From lines 469 to 474.

Whole population vs reference population.

All the parameters are influenced by the definition of founders in the genealogical information based on Pedigree.

When genomic data is used, the founders depend on the number of markers used in the analysis.

Certainly, the reference population as a data set is based in highly reliable pedigree records although we are aware that its depth is not enough in order to properly estimate both ∆F and Nei. These limitations, the need to improve information, as well as the usefulness of the new genomic methodologies have been mentioned throughout the manuscript:

1.- In the “Discussion” section:

“The tendency to increase inbreeding (F) and differences entre las both "berrenda" cattle breeds might also be determined by the different phenotypic (and genotypic selection actions carried out for the eradication of recessive MC1R locus alleles in BN and the eradication of the 1;29 Robertsonian translocation (t1:29) [14,21] in both breeds, and on the other hand, a consequence of the increase in the quality of pedigree information. It should be noted that if higher inbreeding values were estimated in more recent periods due to an improvement in pedigree information, this would lead to overestimate the annual increase in inbreeding rate (Figure 9).”

“For all these and future actions, the current information provided by pedigrees of "berrenda" breeds would be sufficient and should be completed in the short term with the information provided by genomic tests that would allow monitoring genetic variability (such as ROH) and identifying genes linked to traits of economic interest, to disease resistance or to climate change.”

2.- In the “Conclusions” section:

“In this work, we have analyzed for the first time the genealogy of the two Spanish autochthonous “berrenda” cattle breeds, whose analyses provided very useful information not only to the conservationists dealing with these breeds but also to ANABE breeders. It would be necessary the implementation of suitable institutional financial support that allowed ANABE not only to continue with the adequate management of pedigree information, but also to apply genomic methodologies to achieve the conservation objectives and address future selection actions in the "berrenda" cattle breeds.”

Minor comments:

1/ From lines 48 to 52. Please add a reference.

According to reviewer’s indications, a new paragraph has been added:

 “Regarding their phenotypes (Figure 1), both above-mentioned breeds are called “Berrenda” for their coat color pattern. “Berrenda” means “spotted coat” in Spanish language. Although, their morphology [10] and coat color pattern [11,12] are quite similar, they mainly differ in the color of the spots”.

2/ From lines 54 to 55. Figure 1. Please insert photos where the morphology of the breeds could be appreciated.

We have inserted some pictures in order to show the typical morphology of the “berrenda” breeds. In fact, Figure 1 has been substituted by a picture of every “berrenda” breed. Also we have added some others, in order to report the tourism, cultural and ecological interest of these breed. As supplementary materials, we have included a link to a couple of pictures: Figure S1 shows some “berrenda” herds grazing in the “Dehesa” ecosystem and Figure S2 shows some of the ox training

3/ From lines 63 to 67. Please add a reference. Please give more support to this idea. How do authors understand that both breeds maintain the ecosystem equilibrium?

This idea has been supported by adding some new references and also including the next paragraph in the “Introduction” section:

These cattle are reared under extensive conditions, grazing natural meadows and playing an important role in maintaining the “Dehesa” ecosystemic balance by contributing to the maintenance of its vegetation covering [16,17] and its socioeconomic competitiveness [18,19]. Under these breeding conditions and due to their high rusticity, “berrenda” females are excellent mothers for rearing F1 cross-breeding calves, after mating with Charolais or Limousin sires.

4/ From lines 89 to 94. Please add a reference. Please give more details to support this idea.

We are really sorry about this misunderstanding.

GVs have only been calculated in animals belonging to herds whose owners have voluntarily decided to carry out performance controls both in farms and testing centres. But controls are not compulsory to be performed and this issue is not the aim in any of the breeding programs. To be more precise, EBVs have been calculated in some animals but selection measures are not being implemented in order to increase meat production at breed level. Next year, it is intended to start performing artificial insemination in both breeds, and according to that, it would be fundamental to consider the implementation of measures addressed to preserve Genetic Variability in these two breeds.

In order to clarify all this, we have rewritten the next paragraph in the “Introduction” section as follows:

“Moreover, ANABE is planning to implement a program in coming years addressed to improve females rearing skills by analyzing calves’ growing rates and by calculating their estimated breeding value (EBV) using BLUP, the genetic estimation method according to the algorithm by Meuwissen et al. [24]. Semen collection and freezing (for in vitro conservation purposes) are being carried out from those bulls with the highest genetic variability, no t1:29 carriers and in some cases, with their corresponding EBVs. Also, ANABE is planning to start an artificial insemination (AI) program in 2022 addressed to preserve Genetic Variability.”

5/ From lines 91 to 92. Authors mention, “Semen collection and freezing for in vitro conservation purposes are being carried out from bulls with the highest GV”. For genetic conservation, this strategy could bias the variability represented in the semen bank. Did the geneticist take into account the Bulmer effects?  What happens with the other unmeasured traits in the breeds?

We fully agree with reviewer’s comment on this issue and in fact, we think that it will be necessary to think about it at the moment of the implementation of artificial insemination on the “berrenda” herds. In any case, we are aware that the GV of sires should not be the only criteria to be considered. To show all this, this next paragraph has been included at the end of the “Discussion” section:

“The choice of sires and dams to provide germplasm for in vitro conservation, as well as the selection initiatives to eliminate animals carrying t 1:29 disorder or undesirable alleles of the MC1R gene should be carried out by applying optimal contribution selection approaches, in order to achieve these objectives without increasing individual inbreeding.

For all these and future actions, the current information provided by pedigrees of "berrenda" breeds would be sufficient and should be completed in the short term with the information provided by genomic tests that would allow monitoring genetic variability (such as ROH) and identifying genes linked to traits of economic interest, to disease resistance or to climate change.”

6/ From lines 102 to 110. Please add a reference. Please give more details to support this idea. Both paragraphs must be re-written.

According to reviewer’s piece of advice, we have rewritten these paragraphs as follows:

“Although the first registrations of animals in BN and BC studbooks took place in 2001, the traditional rearing system under extensive conditions, in combination with the insufficient control of mating by breeders made quite difficult to correctly assign the paternity to the offspring, what negatively affected to the amount of information provided by pedigrees, in the first stages of the breeding program [35]. Since then, to now, 232 BC herds and 152 BN herds have being registering animals in their corresponding studbooks, reaching 12,057 and 20,389 registered animals in BN and BC, respectively [3].”

7/ From lines 142 to 152. Please clarify the objectives of the study. Look in some way something like considerations or conclusions.

Regarding this, the aims of this work have been clarify as follows:

“As a contribution for the development of the conservation programs in both “berrenda” cattle breeds, this work aims to study the current genetic variability and the structure of BN and BC populations based on thorough analyses of their corresponding pedigrees, by quantifying the changes that happened over the time and comparing them to other different autochthonous cattle breeds that have also been traditionally reared in the “Dehesa” ecosystems, following a quite similar farming system.

To be more precise, the objective of this work are: i) characterizing the structure of BN and BC populations by ages and sex and type of herd; ii) identifying the genetic variability level and the threat of inbreeding in both populations; iii) identifying the genes origin in both populations. It is also intended to know the possible loss of genetic.”

8/ From lines 155 to 158. This sentence looks like the one in lines 111 to 113. Please avoid repeating information.

According to this, the sentence from lines 155 to 158 has been removed from the manuscript.

9/ From lines 171 to 174. Please add a reference.

A new reference by James (1977) has been included in the manuscript and added to the “References” section:

i) Generation interval: It is defined as the average age of parents (male or female) when its replacement is born [47]; it is calculated considering the four possible paths: sire-daughter, sire-son, dam-daughter and dam-calf;”.

10/ From lines 176 to 179. Please add a reference.

A new reference by Maignel et al. (1996) has been included in the manuscript and added to the “References” section:

“iii) Pedigree depth: It is calculated by considering the equivalent number of discreet generations equals to , where n is equal to the number of generations between the individual and its known ancestor; the individuals without known ancestors were assigned to the base generation [49].”.

11/ Additional questions: did the authors consider that marginal genetic contribution estimated by PEDIG software could be relevant in this study? If yes, please incorporate it. If it is not, please widely argue why not.

As far as we are concerned, marginal genetic contributions regarding founder animals and herds were calculated according to ENDOG software, following the same approach as PEDIG software (Boichard, 1997). Marginal genetic contribution results were included in Table 6 of the manuscript. By mistake, this aspect was not mentioned in the methodology. Now, we have included it in the “Material and Methods” section as follows:

“It also considers the genetic variability provided by an animal that could not be explained by the contribution of its offspring. It was calculated according to Boichard [59].”

And also, the next footnote has been added to Table 6:

* The five main ancestors and their genetic marginal contribution identified according to Boichard et al. [59].

12/ From line 259 to 261. Here authors mention something about genomic data. When pedigree is not available or the completeness is not suitable for pedigree analysis. Genomics data could be used instead of pedigree. With the advantage that when genomics are used we are describing the realized genetic variation and not the expected. These arguments must be clarified when genomics data are mentioned in the manuscript.

Reviewer’s indications have been considered and we have tried to clarify the use of genomics but also we have made some comments about the need to keep on working using pedigrees in small livestock breeds throughout the manuscript:

1/ In the “Introduction” section, this the new paragraph has been added:

“Traditionally, performing pedigree analyses was the first step towards the characterization of the genetic diversity within a population but not many works of this kind related to small populations have been published yet [28]. On the other hand, these analyses would also provide a better knowledge of the history of the populations making possible to detect different events from the past that could have affected them, such as the identification of founders’ population or the identification of bottlenecks [29]. Certainly for these purposes, information provided by modern genomic methods is quite better than information provided by pedigree analyses [30-32].  Genomic data would allow for a more accurate assessment addressed to determine different parameters about the structure of the population, such as the Effective population size or the Inbreeding rate [26,27,33], with the added advantage that when using genomic parameters, we would be describing the realized genetic variation and not the expected one. Genomic tools are also quite effective for the conservation of diversity in livestock breeds when they are applied to the study of the genomic linkage disequilibrium due to genetic (selection, mutation, genetic drift, non-random mating…) and non-genetic forces [34]. Genomic analyses demand high financial cost, so the usage of genomic data is not so common in small livestock breeds and analyses based on pedigree data are still the preferential option for monitoring livestock conservation programs, supported by a combination of traditional [35-37] and genomic selection strategies [26,38].”

2/ In the “Discussion” section:

“But it is necessary to be careful in the control of the mating strategies; on the contrary, there would be a high risk of mating between animals belonging to the same lineage, what would lead to an increase of inbreeding [70]. High L values would lead to lower genetic progress and lower genetic gains (in terms of selection of desirable traits) [71]. Genomic selection allows carrying out selective actions addressed to improve some traits of economic interest on the basis of small reference population sizes and moderately long generation intervals, without finding significant differences in the expression of Bulmer’s effect between genomic selection and other traditional selection approaches based on BLUP methodology [72].”.

“For all these and future actions, the current information provided by pedigrees of "berrenda" breeds would be sufficient and should be completed in the short term with the information provided by genomic tests that would allow monitoring genetic variability (such as ROH) and identifying genes linked to traits of economic interest, to disease resistance or to climate change.”.

13/ Line 298. Here authors use reproductive structure. I am not sure but can I assume that the authors refer to population structure?

Assumptions made by reviewer #2 are correct and according to them, the title corresponding to this section has been changed to: “3.1. Population structure and pedigree completeness level”.

14/ I suggest that all the results and discussions be re-written to clarify the presentation of the results.  Please try to argue properly all the concepts.

Following reviewer’s indications to clarify the presentation of the results, we have rewritten the “Results and Discussion” section by removing any unnecessary information and reordering the contents.

Reviewer 3 Report

The present work shows a study on pedigree data on a local population. It is a quite “old” approach using the wide used software Endog but the author gives a complete summary of population parameters and the literature revision and discussion is very complete and well written.

English should be revised because there are several parts that are not correct by syntactical and grammatical point of view and in a general point of view the readability of the manuscript is low. For example, at line 68 “natural character” probably the authors mean temperament or something similar. This is in particular for the introduction part that also in my opinion, should be simplified made more clear.

Line 50 report: “So, while BN shows spots in 50 black, BC is red spotted” that is a repetition of information just reported previously

Line 45-55: Different Herd Book exist for these breeds as indicated in the next paragraphs: in Spain, considered as genetically different entity or if (as appear in the photo) they are reared jointly? And (Line 58) why the colour pattern needs to be fixed?  It seems that the two variety are in process of definition…Please explain better.

Line 90: more correctly indicate the “genetic value” as Estimated breeding value (EBV).

Line 95-123: this part is very long and don’t give essential information related to the work that is going to be presented. Please make a synthesis of these information and reorder the kind of information

Line 247: clue?

Line 271-274: this part is not clear

Line 283: E+ allele is not responsible of black colour in cattle but of agouty or grey coat (Brown Swiss) while black colour is due the ED allele.

Line 439: change Transboundary to transboundary

Figure 8: please make the figure legend more detailed and precise. For example, BN_AR_M could not be clear to the reader that is “average relatedness for male berreda en negro”

Figure 9: what is “J_”?

Line 598 and 599: 1:29 and not 1;29

Table 6: table should be self-explanatory so please avoid abbreviation when it is possible or report even when it is done in the text) the abbreviation in the legend below the table

Line 817: maybe “mating” instead of “mattings”?

Line 818: commercial not Commercial

Line 819: official instead of Official

Line 827: change ; for :

Author Response

RESPONSE Reviewer#3

Review Report Form

Open Review

(x) I would not like to sign my review report

( ) I would like to sign my review report

English language and style

(x) Extensive editing of English language and style required

( ) Moderate English changes required

( ) English language and style are fine/minor spell check required

( ) I don't feel qualified to judge about the English language and style

Yes

Can be improved

Must be improved

Not applicable

Does the introduction provide sufficient background and include all relevant references?

( )

( )

(x)

( )

Is the research design appropriate?

(x)

( )

( )

( )

Are the methods adequately described?

(x)

( )

( )

( )

Are the results clearly presented?

( )

(x)

( )

( )

Are the conclusions supported by the results?

(x)

( )

( )

( )

Comments and Suggestions for Authors

The present work shows a study on pedigree data on a local population. It is a quite “old” approach using the wide used software Endog but the author gives a complete summary of population parameters and the literature revision and discussion is very complete and well written.

1/ English should be revised because there are several parts that are not correct by syntactical and grammatical point of view and in a general point of view the readability of the manuscript is low. For example, at line 68 “natural character” probably the authors mean temperament or something similar. This is in particular for the introduction part that also in my opinion, should be simplified made more clear.

The manuscript has been fully checked in order to improve the English syntax and grammar. Proposed changes have been done, among others.

2/ Line 50 report: “So, while BN shows spots in 50 black, BC is red spotted” that is a repetition of information just reported previously

In order to avoid any repetition of information, this line has been removed from the manuscript.

3/ Line 45-55: Different Herd Book exist for these breeds as indicated in the next paragraphs: in Spain, considered as genetically different entity or if (as appear in the photo) they are reared jointly? And (Line 58) why the colour pattern needs to be fixed?  It seems that the two variety are in process of definition…Please explain better.

BN and BC breeds are genetically and officially recognized as different entities. Coat color needs to be fixed in both  breeds because it is a fundamental feature to phenotipically establish in a clearlier way the singularity and identity of any of these breeds; even more, considering that for a long period of time both breeds were not clearly differentiated of each other. We have tried to clarify this issue by including the next paragraph:

Although the origin of the “berrenda” breeds is considered as ancestral and previous to the Columbian era [4], the first references, published in the 19th century, considered both breeds as a unique racial group [5]. Mitochondrial mt-DNA analyses have shown that both ‘“berrenda’” breeds have different origins [6] and it is being estimated that the separation between them only happened 180 years ago [7]. Consequently, they are officially considered as two distinct breeds, they are subjected to their own specific breeding programs and they have separated studbooks [8] that are managed by ANABE, Group of Berrenda en Negro and Berrenda en Colorado Cattle Breeders Associations [9].

Regarding their phenotypes (Figure 1), both above-mentioned breeds are called “Berrenda” for their coat color pattern. “Berrenda” means “spotted coat” in Spanish language. Although, their morphology [10] and coat color pattern [11,12] are quite similar, they mainly differ in the color of the spots. So, while BN shows spots in black, BC is red spotted. Similarly to other local breeds [13], coat color and spot patterns in both “berrenda” cattle breeds are relevant for their appearance and for the differentiation between them, especially when they are used in traditional events. In order to achieve the fixation of black coat color in BN and to avoid births of animals with different coat color that could not be admitted as BN breeding animals, BN studbook in combination with genotyping of MC1R locus have been used [14].”

4/ Line 90: more correctly indicate the “genetic value” as Estimated breeding value (EBV).

According to the previous comment made by reviewer #3, we have corrected the text and the paragraph has been rewritten as follows:

“Moreover, ANABE is planning to implement a program in coming years addressed to improve females rearing skills by analyzing calves’ growing rates and by calculating their estimated breeding value (EBV) using BLUP, the genetic estimation method according to the algorithm by Meuwissen et al. [24]. Semen collection and freezing (for in vitro conservation purposes) are being carried out from those bulls with the highest genetic variability, no t1:29 carriers and in some cases, with their corresponding EBVs. Also, ANABE is planning to start an artificial insemination (AI) program in 2022.”

5/ Line 95-123: this part is very long and don’t give essential information related to the work that is going to be presented. Please make a synthesis of this information and reorder the kind of information

Regarding the previous suggestions, we have reorganized, clarified and summarize this part of the manuscript in the following way:

“Although the first registrations of animals in BN and BC studbooks took place in 2001, the traditional rearing system under extensive conditions, in combination with the insufficient control of mating by breeders made quite difficult to correctly assign the paternity to the offspring, what negatively affected to the amount of information provided by pedigrees, in the first stages of the breeding program [35]. Since then, to now, 232 BC herds and 152 BN herds have being registering animals in their corresponding studbooks, reaching 12,057 and 20,389 registered animals in BN and BC, respectively [3].

In the case of “berrenda” cattle breeds, genetic diversity and paternity of the offspring are annually monitored by analyzing DNA microsatellites, according to FAO recommendations [39]. Although the neutral genetic variability is keeping reasonably high in both “berrenda” breeds, apparently it is decreasing in the last 6 years, as a consequence of the implementation of the above-mentioned actions [40], what could make necessary to complete the studies on genetic variability by using a different methodology.”

6/ Line 247: clue?

This line has been removed after considering some indications made by reviewer #2.

7/ Line 271-274: this part is not clear

These lines have been removed with the intention of reducing not relevant or repeated information.

8/ Line 283: E+ allele is not responsible of black colour in cattle but of agouty or grey coat (Brown Swiss) while black colour is due the ED allele.

We fully agree with this comment, so this sentence has been changed as follows:

“ii) the lack of fixation of MC1R locus dominant allele ED in BN population, responsible of black coat color in BN what could lead to the feasible birth of red or chestnut coat color calves which cannot be registered in the BN studbook nor used as breeding animals [3]”.

9/ Line 439: change Transboundary to transboundary

The previous correction has been made.

10/ Figure 8: please make the figure legend more detailed and precise. For example, BN_AR_M could not be clear to the reader that is “average relatedness for male berrenda en negro”

Following reviewer’s advice, we have modified the legend of this figure in the following way:

Figure 8. Average relationship (AR) coefficients inbreeding rates (%) in herds, males (M) and females (F) from reference populations of “Berrenda en Negro” (BN) and “Berrenda en Colorado” (BC) cattle breeds.”

11/ Figure 9: what is “J_”?

In order to clarify, we have modified the legend of this figure in the following way:

Figure 9. Average inbreeding values (F), average individual increase of inbreeding (∆F), and average equivalent generations (GenEqu) in the “Berrenda en Negro” (BN) and “Berrenda en Colorado” (BC) studbooks per year of birth of the individuals belonging to the reference populations.”

12/ Line 598 and 599: 1:29 and not 1;29

The previous correction has been made.

13/ Table 6: table should be self-explanatory so please avoid abbreviation when it is possible or report even when it is done in the text) the abbreviation in the legend below the table

The previous correction has been made.

14/ Line 817: maybe “mating” instead of “mattings”?

The previous correction has been made throughout the manuscript.

15/ Line 818: commercial not Commercial

The previous correction has been made.

16/ Line 819: official instead of Official

The previous correction has been made.

17/ Line 827: change ; for :

The previous correction has been made.

Round 2

Reviewer 1 Report

The manuscript has been significantly improved by the authors and can be published in this form

Reviewer 2 Report

Accept in present form

Reviewer 3 Report

Dear auhtors , thank you very much for accepting the correction. In my opinion the manuscript could be published as is.